# ONLINE DECISION MAKING WITH GENERATIVE ACTION SETS

**Jianyu Xu, Vidhi Jain, Bryan Wilder, Aarti Singh**
Carnegie Mellon University
Pittsburgh, PA 15213
{jianyux, vidhij2, bwilder, aarti}@andrew.cmu.edu

## ABSTRACT

With advances in generative AI, decision-making agents can now dynamically create new actions during online learning, but action generation typically incurs costs that must be balanced against potential benefits. We study an online learning problem where an agent can generate new actions at any time step by paying a one-time cost, with these actions becoming permanently available for future use. The challenge lies in learning the optimal sequence of two-fold decisions: which action to take and when to generate new ones, further complicated by the triangular tradeoffs among exploitation, exploration and *creation*. To solve this problem, we propose a doubly-optimistic algorithm that employs Lower Confidence Bounds (LCB) for action selection and Upper Confidence Bounds (UCB) for action generation. Empirical evaluation on healthcare question-answering datasets demonstrates that our approach achieves favorable generation-quality tradeoffs compared to baseline strategies. From theoretical perspectives, we prove that our algorithm achieves the optimal regret of $O(T^{\frac{d}{d+2}} d^{\frac{d}{d+2}} + d\sqrt{T \log T})$, providing the first sublinear regret bound for online learning with expanding action spaces.

## 1 INTRODUCTION

Sequential decision-making problems involve agents repeatedly selecting actions from a candidate set to maximize cumulative reward. Traditional approaches assume a fixed set of available actions, focusing on the exploration-exploitation tradeoffs: balancing empirically high-reward actions (exploitation) against less-tested alternatives (exploration). However, advances in generative AI have introduced a new paradigm where contemporary systems can dynamically *expand* their action spaces by *creating* novel actions over time. This capability introduces an additional strategic dimension that agents should also balance immediate performance with strategic investments in future capabilities enabled by new actions. Consider the following motivating scenarios:

**Example 1.1** (Healthcare Question-Answering Systems). *AI-powered healthcare platforms must decide between reusing existing vetted responses from their FAQ libraries or investing in creating new, tailored responses for novel patient inquiries. Each custom response requires costly expert review and validation (potentially hundreds of dollars when accounting for clinical expertise). However, once created and vetted, these responses become reusable assets. When a patient in a given region asks "What are healthy meals during pregnancy?", the system faces a critical choice: provide a generic response about pregnancy nutrition, or invest in creating a new response more specific to typical foods in that region, benefiting hundreds of future expectant mothers in similar settings.*

**Example 1.2** (Personalized Advertisement). *An advertising platform may initially start with a finite set of ad templates for different user contexts. Over time, the platform observes new user segments and decides to design specialized ads (with initial design and production costs) perfectly customized to the new user subgroups. Once created, these specialized ads become available for future targeting at no additional cost.*

In both scenarios, the agent must decide at each time step whether to select an existing action or pay a one-time cost to instantiate a new action perfectly suited to the observed context. This introduces a novel *create-to-reuse* problem that goes beyond traditional exploration-exploitation tradeoffs.

**Problem Formulation (preview).** We study an online learning problem with an *expanding* action space. At each time $t$, the agent first observes a context $x_t$. Then it may either

(a) Select an *existing* action, incurring some (potentially suboptimal) loss, or

(b) *Generate* a new action that is customized the current $x_t$ (without any excess loss), at a fixed one-time cost.

This formulation has two key features. First, step (b) is notable in that the agent generates a new action only through an oracle $\mathcal{A}(x_t)$ that is prompted by the context $x_t$. The learning algorithm operates as a decision-making layer on top of this custom action oracle. In contrast, traditional online learning or bandits operate directly in the action space. Second, once generated, new actions can be reused in future rounds without additional expense. The key is to judiciously decide when to pay the cost of adding such a specialized action and when to rely on existing arms.

This setting presents fundamental challenges that distinguish it from existing online learning and bandits frameworks. We face a triangular tradeoffs among three competing objectives: exploitation (using known good actions), exploration (learning about existing uncertain actions), and **creation** (generating a new action to satisfy immediate needs while enriching future capabilities). Additionally, we have no prior experience with potential new actions or unlimited freedom to generate arbitrary ones – each creation must be specifically tailored to the current context at a fixed cost.

**Summary of Contributions**  Our main contributions are fourfold:

1. **Problem Modeling**: We establish a new problem formulation that allows for costly expansion of the action space in online learning, formalizing the create-to-reuse framework.

2. **Algorithmic Framework**: We propose a *doubly-optimistic* algorithm that uses Lower Confidence Bounds (LCB) when selecting among existing actions, and Upper Confidence Bounds (UCB) when deciding whether to generate new actions. This design simultaneously exploits near-optimal actions and enables creation without excessive hesitation.

3. **Empirical Validation**: We conduct experiments on real-world healthcare question-answering datasets, demonstrating that our approach achieves favorable generation-quality tradeoffs compared to baselines. Our results show the method gracefully interpolates between pure reuse and always-create policies while maintaining superior performance.

4. **Optimal Regret Guarantees**: Under a semi-parametric loss model, our algorithm achieves $O(T^{\frac{d}{d+2}} d^{\frac{d}{d+2}} + d\sqrt{T \log T})$ expected regret, where $T$ is the time horizon and $d$ is the dimension of covariates. We prove this rate is optimal by establishing a matching $\Omega(T^{\frac{d}{d+2}})$ information-theoretic lower bound.

**Technical Novelty.** The crux of our approach is a **double optimism** principle, which resolves the unique challenge of balancing creation with exploration/exploitation. Among existing actions, we rely on their *LCB* comparisons to both exploit high-performing actions and continue exploring uncertain ones. When evaluating creation decisions, we compare the *UCB* loss of the best existing action against the fixed generative cost, triggering creation with appropriate probability. This double optimism perspective naturally maximizes the long-term value of new actions while tightly controlling worst-case regret.

**Paper Organization.** The rest of this paper is organized as follows. We discuss related literature in Section 2, then present the rigorous problem setting in Section 3 along with the necessary assumptions. We propose our main algorithm in Section 4, analyze its theoretical performance in Section 5, and conduct numerical experiments to validate its real-world performance in Section 6.

## 2 RELATED WORKS

Here we discuss related literature on the most relevant topics in online decision making. Please refer to Appendix A for discussions on broader fields including bandits with knapsacks, active learning, and digital healthcare.

**Multi-Armed and Contextual Bandits.** The multi-armed bandit (MAB) problem has been extensively studied since Lai & Robbins (1985). The classic framework (Auer et al., 2002; Agarwal et al., 2014), that a decision-maker repeatedly selects from a fixed set of arms, was extended to contextual bandits (Li et al., 2010; Chu et al., 2011) where rewards depend on observable contexts. The crux is to balance exploration and exploitation with the goal of *regret* minimization. Please refer to Slivkins et al. (2019) for a comprehensive discussion.

**Online Facility Location.** Online facility location (OFL), studied by Meyerson (2001), Fotakis (2008), and Guo et al. (2020), is closely related to our formulation. In OFL, algorithms decide whether to open new facilities or assign requests to existing ones, minimizing facility costs plus assignment distances. While structurally similar to our problem, there are crucial differences. First, OFL assumes *known* distance metrics, while we must learn *unknown* parameters defining distances. Second, OFL *automatically* assigns points to nearest facilities, while we must *actively* select actions under uncertainty. Therefore, OFL involves a *two-way* trade-offs between immediate costs and future benefits, whereas our problem requires a *three-way* balance between exploitation, exploration, and creation, necessitating our novel algorithmic approach.

## 3 PROBLEM SETUP

We now formalize the problem of creating-to-reuse as an online decision-making framework. In order to demonstrate the problem setting, we start with the healthcare Q&A scenario described in Example 1.1. As an abstraction, each arriving patient question is represented as a $d$-dimensional *context* vector $x_t$ in a learned semantic embedding space. The system maintains a context library $S_t$ of vetted FAQ entries, implemented as a *hash table* where each context that has been previously added serves as a key to its corresponding custom respond (or generally the *action*) generated by an oracle $\mathcal{A}(\cdot)$. Crucially, the algorithm operates only in the context representation space by searching through context keys in $S_t$. When a new question $x_t$ arrives, the algorithm makes decisions based on estimated losses and can either:

(a) Decide to create a new custom response by paying a fixed cost $c$ and adding context $x_t$ as a new key to the library. The generation oracle $\mathcal{A}(\cdot)$ then automatically produces the tailored action $a_t = \mathcal{A}(x_t)$, and the pair $(x_t, a_t)$ becomes permanently available for future reuse. *Or*

(b) Select an existing context key $f \in S_t$ from the library. The system automatically retrieves the corresponding action $a_t = S_t(f) = \mathcal{A}(f)$ and deploys it for context $x_t$, incurring a mismatch loss $d(x_t, f)$ that reflects the difference between (1) the custom response to context $x_t$ versus (2) the action tailored for another context $f$.

Technically, we consider the following problem setting.

---

Initialize: A library $S_1 = \{f, \mathcal{A}(f)\}$ with context keys $f$ and vetted custom actions $\mathcal{A}(f)$.
For $t = 1, 2, ..., T$ :

1. Observe $x_t \in \mathbb{R}^d$ (patient question arrives).
2. The algorithm decides whether to create a customized response to $x_t$. If YES, then
    (i) Generation oracle produces and deploys $a_t = \mathcal{A}(x_t)$ (custom response to $x_t$).
    (ii) Receive a fixed loss $c$ (creation cost).
    (iii) Update $S_{t+1} := S_t \cup \{x_t : a_t\}$ (add new context-action pair to the library).
3. If NO, then
    (i) Select an existing context key $f_t \in S_t$ and retrieve $a_t = S_t(f_t)$.
    (ii) Receive a loss $l_t := d(x_t, f_t) + N_t$ (noisy mismatch penalty).
    (iii) Update $S_{t+1} := S_t$ (library unchanged).

---

In this formulation, $d(x_t, f_t)$ captures the expected mismatch loss when deploying an action originally designed for context $f_t$ to serve context $x_t$. While this fundamentally reflects the difference between $\mathcal{A}(x_t)$ and $\mathcal{A}(f_t)$ in the action space, the algorithm can only estimate this through context-space relationships since it lacks direct access to $\mathcal{A}(x_t)$ (actions having not been generated yet).

For theoretical analysis, our main modeling assumption is that this mismatch can be captured by a *squared distance* function in the context space. In experiments, we consider other forms for the mismatch distance.

**Assumption 3.1** (Quadratic parametric loss)**.** We assume the distance function satisfies

$$d(x, f) := (x - f)^\top W (x - f) \tag{1}$$

where $W \in \mathbb{S}_+^d$ is an **unknown** positive semi-definite $d \times d$ matrix. Accordingly, denote

$$
\begin{aligned}
w &:= Vec(W) \in \mathbb{R}^{d^2} \\
\phi(x, f) &:= Vec[(x - f)(x - f)^\top] \in \mathbb{R}^{d^2},
\end{aligned}
\tag{2}
$$

and we have an equivalent definition as $d(x, f) := \phi(x, f)^\top w$.

**Why we assume a quadratic parametric loss?** The motivation is that contexts are embedded in a space where different dimensions capture semantically relevant information. The cost of reusing an action designed for one context when serving another can be modeled as a distance on this representation space, although the exact importance weighting of different semantic dimensions (captured by matrix $W$) is unknown to the learner. Since $d(x_t, x_t) = 0$, our formulation measures the *excess* cost due to not generating a custom action for each $x_t$. This fits scenarios where the algorithm interacts with complex action spaces through oracle $\mathcal{A}(x_t)$ (human expert or generative model); our aim is to achieve good performance relative to this oracle's capabilities. Modeling $d(\cdot, \cdot)$ as a squared distance function captures more structure than linear parametric choices while remaining more tractable than nonparametric formulations. Furthermore, the empirical results our algorithm performs on real-world Healthcare Q&A datasets validate the robustness of this modeling.

**Goal of Algorithm Design.** Our goal is to minimize the expected *total loss*. We will rigorously define the performance metric and technical assumptions at the beginning of Section 5.

## 4 ALGORITHM

To solve this online decision-making problem with expanding context libraries, we propose a "Doubly-Optimistic" algorithm. This section presents the algorithm design and highlights its properties. We will analyze and bound its cumulative regret in the next section.

The pseudocode of our algorithm is displayed in Algorithm 1. At each time $t$, the algorithm inherits loss estimation parameters $\Sigma_{t-1}$, $b_{t-1}$ [1] and context library $S_t$ from $(t - 1)$, then observes a new context vector $x_t$. Using the estimation parameters, it computes predicted mismatch loss $\bar{d}_t(x_t, f)$ and uncertainty bound $\Delta_t(x_t, f)$ for each existing context key $f \in S_t$. The algorithm operates entirely in the context representation space and takes the following two steps to determine which action to deploy.

(i) **Lower Confidence Bound (LCB) loss on existing contexts.** For each existing context key $f$, we calculate the LCB loss as $\check{d}_t(x_t, f) = \bar{d}_t(x_t, f) - \Delta_t(x_t, f)$. We then select $f_t$ as the context key with the minimum LCB loss. Note that we do not immediately choose to reuse this context.

(ii) **Upper Confidence Bound (UCB) probability to create a new entry.** After identifying $f_t$ as the best existing option, we turn to consider its UCB loss $\hat{d}_t(x_t, f_t) = \bar{d}_t(x_t, f_t) + \Delta_t(x_t, f_t)$ and compare it against the fixed creation cost $c$. With a probability of $\min\{1, \frac{\hat{d}_t(x_t, f_t)}{c}\}$, we decide to create a new entry: The oracle generates $a_t = \mathcal{A}(x_t)$ and we add context $x_t$ to the library. Otherwise, we reuse the existing context $f_t$, and the system retrieves $a_t = S_t[f_t]$ from library $S_t$. After receiving a mismatching loss $l_t$, the algorithm update the estimation parameters $\Sigma_t$ and $b_t$ accordingly.

As the argmin of LCB loss, $f_t$ represents the existing context that could potentially yield the lowest mismatch under optimistic assumptions, balancing exploration and exploitation given historical uncertainties. This approach aligns with contextual bandit methods such as Chu et al. (2011).

---

[1] Linear regression parameters. We estimate the vector $w$ as $\Sigma_{t-1}^{-1} b_{t-1}$ at every time step $t$.

---

**Algorithm 1** Doubly-Optimistic Algorithm

---

1: **Initialization: Custom action oracle** $\mathcal{A}(x)$, and $\Sigma_0 = \lambda \cdot I_{d^2}, b_0 = \vec{0}, S_1 = \{\vec{1}_d : \mathcal{A}(\vec{1}_d)\}, \alpha$.
2: **for** $t = 1, 2, \ldots, T$ **do**
3:     Observe context $x_t \in \mathbb{R}^d$
4:     **for** $\forall f \in S_t$ (all existing context keys) **do**
5:         **Compute loss estimates and uncertainties**. Denote

$$\Delta_t(x, f) := \alpha \cdot \sqrt{\phi(x_t, f)^\top \Sigma_{t-1}^{-1} \phi(x_t, f)}, \quad \bar{d}_t(x, f) := \phi(x, f)^\top \Sigma_{t-1}^{-1} b_{t-1}$$
$$\hat{d}_t(x, f) := \bar{d}_t(x, f) + \Delta_t(x, f), \qquad \check{d}_t(x, f) := \bar{d}_t(x, f) - \Delta_t(x, f) \tag{3}$$

6:     **end for**
7:     Select context key $f_t := \arg\min f \in S_t \check{d}_t(x_t, f)$.
8:     **if** $Z_t == 1$ with $Z_t \sim Ber(\min\{1, \frac{1}{c} \cdot \hat{d}_t(x_t, f_t)\})$ as an i.i.d. Bernoulli random variable **then**
9:         **Decide to create new**: Oracle generates action $a_t = \mathcal{A}(x_t)$ and deploys it at a cost $c$.
10:         Receive loss $l_t = 0$ (perfect match for custom action).
11:         Update context library $S_{t+1} = S_t \cup x_t : a_t$ (add new context-action pair).
12:         Keep $\Sigma_t := \Sigma_{t-1}$ and $b_t := b_{t-1}$ without updating.
13:     **else**
14:         **Decide to reuse**: Retrieve and deploy action $a_t = S_t[f_t] = \mathcal{A}(f_t)$ at no creation cost.
15:         Receive mismatching loss $l_t = d(x_t, f_t) + N_t$.
16:         Maintain context library $S_{t+1} = S_t$ (no new entries).
17:         Update loss estimation parameters

$$\Sigma_t := \Sigma_{t-1} + \phi(x_t, f_t)\phi(x_t, f_t)^\top, \quad b_t := b_{t-1} + l_t \cdot \phi(x_t, f_t). \tag{4}$$

18:     **end if**
19: **end for**

---

The UCB-based creation probability $\frac{\hat{d}_t(x_t, f_t)}{c}$ increases the chance of adding new contexts when the estimated mismatching loss is high relative to creation cost (within a risk $\Delta_t$ we can tolerate). This design enables us to estimate the "necessity" of creating new entries while bounding the total expected loss accumulated *before* any new context is added to a particular region of the context space. We explain this property later in Lemma B.5.

**Computational Complexity**      Algorithm 1 incurs worst-case time complexity $O(d^4 T^2)$, as it compute matrix-vector products of $d^2$ dimension for every context key $f \in S_t$ at each round $t$, with at most $T$ contexts possible. Since the expected number of newly created contexts is $O(T^{\frac{d}{d+2}})$ with respect to $T$, the *expected* complexity refines to $O(T^{\frac{2d+2}{d+2}})$. Given that $d$ can represent sentence embedding dimensions in the Q&A scenario, an $O(d^4)$ time complexity is impractical. In our real-data numerical experiments, we improve computational performance by replacing the estimated distance function $\bar{d}_t(x, f)$ with either of the two forms:

(a) A squared linear model $(\theta^\top(x-f))^2$ (equivalent to setting $W = \theta\theta^\top$), reducing the computational complexity to $O(d^2)$. Uncertainty bounds are derived from ridge regression covariance matrices.

(b) A neural network $d(x, f; \Theta)$ with uncertainty function $\Delta_t(x, f; \Theta)$ derived under Gaussian conjugate assumptions, reducing complexity to $O(D^2)$ where $D := \|\Theta\|_0$ is the number of NN parameters.

On the other hand, we implement the original algorithm in the synthetic-data simulations to validate the theoretic guarantees with respect to $T$ (for small $d$'s only).

## 5   THEORETICAL ANALYSIS

In this section, we provide a regret analysis of our algorithm. We first state the performance metric and necessary technical assumptions. Then we present the main theorem on the algorithmic regret

upper bound. Finally, we provide a corresponding lower bound that matches the leading term of the upper bound with respect to $T$.

## 5.1 DEFINITIONS AND ASSUMPTIONS

As we have stated by the end of Section 3, our goal is to minimize the *total loss*. In order to measure the performance, we adopt the expected *regret* as the loss metric, which is defined as follows:

**Definition 5.1** (Optimal and Regret). Denote the minimal expected loss[2] that is *achievable* in hindsight as $OPT_h$, which equals:

$$OPT_h := \min_{\mathcal{S}:=\{S_1,S_2,\ldots,S_T,S_{T+1}|S_{t+1}\setminus S_t \subseteq \{x_t\}\}} c \cdot |S_{T+1}| + \sum_{t=1}^{T} \min_{f \in S_{t+1}} d(x_t, f). \quad (5)$$

There also exists a *non-achievable* minimal loss denoted as $OPT_o$, which is only accessible by an omniscient oracle that knows $\{x_t\}_{t=1}^T$ and selects an optimal option set ahead of time:

$$OPT_o := \min_S c \cdot |S| + \sum_{t=1}^{T} \min_{f \in S} d(x_t, f). \quad (6)$$

From the definition, we know that $OPT_o \leq OPT_h$. Also, denote the expected loss obtained by our algorithm as $ALG$, which equals:

$$ALG := c \cdot |S_{T+1}| + \sum_{t=1}^{T} \min_{f \in S_{t+1}} d(x_t, f). \quad (7)$$

Define the regret $REG$ as the expected loss difference[3] between $OPT_h$ and $ALG$.

$$REG := \mathbb{E}[ALG - OPT_h] \quad (8)$$

We then make two distributional assumptions on the covariates and the noises, respectively.

**Assumption 5.2** (Covariate distribution and norm bound). Assume $x_t \in \mathbb{R}^d, t = 1, 2, \ldots, T$ are drawn from independent and identical distributions (i.i.d.), with $d \geq 2$. Also, assume a norm bound as $\|x_t\|_2 \leq 1$.

Assumption 5.2 is necessary for us to effectively learn the metric matrix $W$ through online linear regression. For the same reason, we assume a subGaussian noise on the observations as follows:

**Assumption 5.3** (Noise distribution). Assume that $N_t \in \mathbb{R}, t = 1, 2, \ldots, T$ are drawn from $\sigma$-*subGaussian i.i.d.*, where $\sigma$ is a universal constant.

## 5.2 REGRET BOUNDS

In this subsection, we sequentially present our theoretical guarantees on the regret upper and lower bounds, as the following two theorems.

**Theorem 5.4** (Regret upper bound). *With assumptions made in Section 5.1, the expected regret of our Algorithm 1 is upper bounded by $O(T^{\frac{d}{d+2}} d^{\frac{d}{d+2}} + d\sqrt{T \log T})$.*

*Proof Sketch.* We prove Theorem 5.4 in the following sequence:

1. (Lemma B.1) We upper bound the non-achievable minimal loss as $OPT_o = O(T^{\frac{d}{d+2}} d^{\frac{d}{d+2}})$. This is proved by a fine-grid covering of the space.

2. (Lemma B.3) We upper bound the algorithmic loss $ALG$ within a constant competitive ratio of $OPT_o$ adding cumulative prediction errors: $\mathbb{E}[ALG] = O(\mathbb{E}[OPT_o + \sum_{t=1}^{T} \Delta_t(x_t, f_t)])$. To prove this, we divide $\{x_t\}$'s into "good" and "bad" groups, and bound their excess loss respectively.

---

[2]Expectation taken over observation noises only. Same for the definition of $OPT_o$.

[3]Expectation taken over the $\{x_t\}_{t=1}^T$ series.

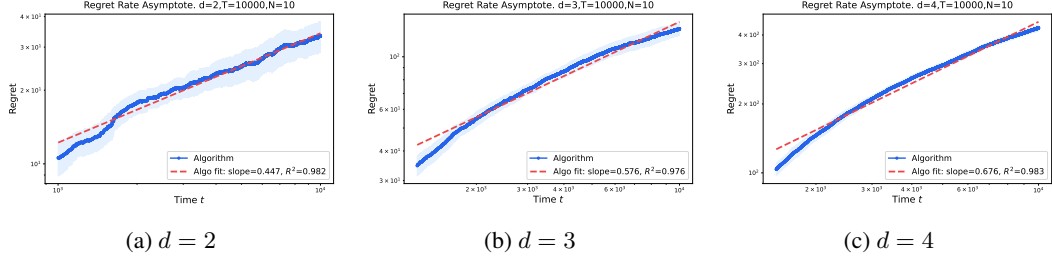

(a) $d = 2$           (b) $d = 3$           (c) $d = 4$

Figure 1: Regret curves for $T = 10000$ and $d = 2, 3, 4$ in log-log scales, repeated by $N = 10$ epochs. The slope of the linear asymptote under log-log diagram indicates the power dependence of regret on $T$, which should be $\frac{d}{d+2}$.

3. (Lemma B.8) We upper bound the excess risk $\mathbb{E}[\sum_{t=1}^{T} \Delta_t(x_t, f_t)] = O(d\sqrt{T \log T})$ by standard online linear regression (similar to Chu et al. (2011) by replacing $d$ with $d^2$).

4. Finally, we derive the regret upper bound as $REG = \mathbb{E}[ALG - OPT_h] = O(T^{\frac{d}{d+2}} d^{\frac{d}{d+2}} + d\sqrt{T \log T})$ according to the three steps above.

Please refer to Appendix B for all technical details of this proof, including rigorous statements of lemmas and derivations of inequalities.          $\square$

To show the optimality of the regret upper bound proposed above, we present the information-theoretic lower regret bound.

**Theorem 5.5** (Regret lower bound). *For any online learning algorithm, there exists an instance of problem setting presented in Section 3, such that the regret is at least $\Omega(T^{\frac{d}{d+2}})$ with respect to $T$ (despite the dependence on $d$).*

We defer the proof to Appendix B.6. The main idea is to apply the $\Omega(K^{-\frac{2}{d}})$ lower bound for the K-nearest-neighbors (K-NN) problem, along with an optimal choice of $K$ that balance this term with $c \cdot K$. Theorem 5.5 indicates that our algorithm achieves an optimal regret with respect to $T$.

## 6 EMPIRICAL PERFORMANCE

In this section, we conduct numerical experiments to validate our method's performance. We first run the original algorithm on low-dimensional synthetic data to demonstrate the regret dependence on $T$. Then we adapt our algorithm to real-world healthcare Q&A scenarios and show better trade-offs between generation cost and mismatching loss compared to baselines.

### 6.1 REGRET VALIDATION ON SYNTHETIC DATA

We evaluate our doubly-optimistic algorithm on synthetic data with dimensions $d = 2, 3, 4$ over time horizon $T = 10,000$, repeated for $N = 10$ epochs. Context vectors $x_t$ are drawn from $L_2$-normalized uniform distributions, with noise $N_t \sim \mathcal{N}(0, 0.05)$. We calculate regret by comparing the algorithmic loss against $OPT_o$ (defined in Equation (6)), approximated by randomized K-means++ with Lloyd iterations over potentially optimal values of $K$. We do not apply $OPT_h$ as its computational cost is exponentially dependent on $T$.

Figure 1 presents the regret curves in log-log scale to reveal the power dependence of regret on $T$. Our method exhibits empirical slopes of $0.447, 0.576, 0.676$ for $d = 2, 3, 4$ respectively, aligning closely with the theoretical rates which should be $\frac{d}{d+2}$ according to Theorem 5.4. These results validate our theoretical analysis in synthetic environments.

Note: We restrict experiments to low-dimensional settings due to the computational cost of $OPT_o$ (a necessary component of regret) in high dimensions, where K-means++ becomes ineffective and the

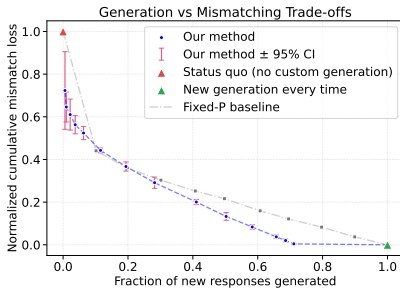 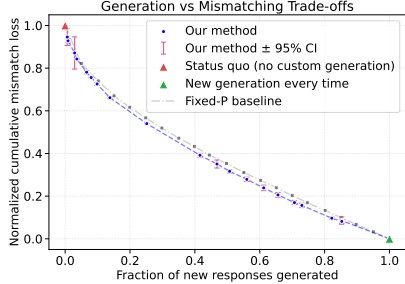

(a) Numerical results on a private maternal health Q&A dataset.

(b) Numerical results on the public Medical Q&A Dataset.

Figure 2: Tradeoffs between normalized generation costs (x-axis) and normalized mismatching loss (y-axis) on two healthcare Q&A datasets. A lower/left curve indicates a better performance. Each blue point represents the (generation cost, mismatching loss) pair caused by a choice of $c$. In both cases, our algorithm outperforms the baseline that randomly generates custom responses with a variety of fixed probabilities $p$ (each gray point represents a choice of $p$).

underlying nearest neighbor problem is NP-hard. Despite these computational limitations, the synthetic validation confirms that our approach achieves the predicted theoretical regret rates, providing confidence in its performance for moderate-dimensional real-world applications.

## 6.2 GENERATION-QUALITY TRADEOFFS ANALYSIS ON HEALTHCARE Q&A DATASETS

We evaluate our algorithm on two real-world healthcare Q&A datasets to demonstrate its practical effectiveness:

1. **Private Maternal Health Dataset**: A dataset containing 839 user queries, with 12 pre-written FAQs for pregnant women, provided by a company developing healthcare chatbots.

2. **Medical Q&A Dataset**: A public collection of 47,457 medical question-answer pairs curated from 12 NIH websites (`https://www.kaggle.com/datasets/gvaldenebro/cancer-q-and-a-dataset`).

**Experimental Setup.** Our experimental framework models the create-to-reuse decision process operating entirely in the context representation space. All questions are mapped to embeddings using OpenAI's pre-trained `text-embedding-3-small` model, creating a semantic representation space where the algorithm makes decisions. For each arriving question context $x_t$, the algorithm decides whether to select an existing context key $f$ from the FAQ library or add $x_t$ as a new context key and then invoke the custom answer generation oracle $\mathcal{A}(\cdot)$.

Custom answer generation differs across datasets to reflect their nature. For the maternal health dataset, custom answers are generated by GPT-5 with carefully designed prompts including safety guardrails and emergency detection protocols appropriate for healthcare contexts. For the Medical Q&A dataset, custom answers are directly retrieved from the pre-existing responses associated with each question entry.

Crucially, the mismatch loss feedback occurs in the action space rather than the context space. For current question context $x_t$ and an existing context $f$ in the FAQ library, the loss is calculated as $(1 - \text{cosine similarity})$ between $x_t$**'s custom answer** and $f$**'s custom answer**. This reflects our core assumption that the algorithm operates in context space while true loss manifests in action space, accessible only through the generation oracle $\mathcal{A}(\cdot)$.

As we also mentioned in Section 4, to maintain computational tractability, we model the estimated loss function as $\bar{d}(x, f) = (\theta^\top (x - f))^2$ (on the maternal health dataset) or adopt a neural network $d(x, f; \Theta)$ (on Medical Q&A Dataset).

We evaluate our doubly-optimistic algorithm against a fixed-probability baseline strategy. This baseline makes i.i.d. Bernoulli decisions $\sim \text{Ber}(p)$ at each time step: with probability $p$, generate a cus-

tom response; otherwise, select the most similar existing context from the library based on cosine similarity between question embeddings (note that it has no access to the custom answer before generation). To comprehensively evaluate performance across different cost-accuracy preferences, we vary the probability parameter $p$ uniformly across $[0, 1]$ for the baseline. Meanwhile, we also vary the creation cost parameter $c$ from 0 to 100 for our algorithm, generating complete tradeoff curves for both approaches.

The numerical results are depicted in Figure 2, where points and curves closer to the bottom-left indicate superior performance. We plot cumulative generation costs against cumulative mismatch losses, with both metrics normalized separately to $[0, 1]$ scale for interpretability. Generation costs are normalized by the total cost of the always-generate strategy, while mismatch losses are normalized by the *status quo* strategy that never generates custom responses. Note that these represent the two components of total loss in our formulation, depicted separately for clearer analysis. Each gray point represents a different choice of $p$ for the baseline, forming a curve that represents the best possible performance achievable by any fixed-probability strategy. Each experiment runs $N = 10$ epochs with 95% confidence intervals computed using Wald's test.

**Results on the Maternal Health Dataset.** Figure 2a presents the generation-quality tradeoffs. Starting with 12 pre-written FAQs, our algorithm demonstrates several key advantages:

1. **Context Clustering**: Compared with the always-generating strategy (green triangle), approximately 30% of user questions exhibit sufficient similarity to existing FAQs, as evidenced by the algorithm achieving near-zero mismatch loss when generating responses for 70% of queries.

2. **Efficiency Gains**: Compared with *status quo* (red triangle), strategic addition of just a few targeted FAQs reduces mismatch loss by approximately 25% (as evidenced by the algorithmic curve approaching the point $(0, 0.75)$), highlighting the value of adaptive creation decisions over static policies.

3. **Pareto Optimality**: Our algorithm consistently outperforms fixed-probability baselines throughout the entire generation spectrum, with statistical significance demonstrated by 95% confidence intervals. The doubly-optimistic approach effectively pushes the performance frontier toward Pareto optimality.

**Results on Medical Q&A Dataset.** Figure 2b presents results on the public Medical Q&A dataset. We establish the initial FAQ library by prompting GPT-5 to classify all questions into 32 categories by topic, then randomly sampling 10 question-answer pairs from each category to create a generic response (a total of 32 FAQs).

Compared with the always-generating and FAQ-only baselines respectively, our algorithm can reduce about 60% generation cost and about 60% mismatch loss, leading to positive-sum tradeoffs (indicated by the convex curve). Also, it achieves statistically significant improvements over fixed-probability baselines across nearly the entire generation spectrum, as confirmed by 95% confidence intervals. However, the performance gains are notably smaller than those observed on the private maternal health dataset. We attribute this difference to the greater diversity in the Medical Q&A dataset, spanning 37 question types across 32 medical topics. In contrast, the private dataset focuses specifically on maternal health with more concentrated topics and frequently recurring keywords, producing clearer semantic connections and stronger correlations between context and action similarities that enable more effective learning.

The results validate our theoretical framework in practice, demonstrating that principled confidence bound approaches for creation decisions significantly outperform heuristic alternatives in real-world healthcare applications where both response quality and resource efficiency are critical.

## 7 CONCLUSION

In this paper, we introduced an online decision-making problem where new actions can be generated on the fly, at a fixed cost, and then reused indefinitely. To address the balance among exploitation, exploration, and creation, we proposed a doubly-optimistic algorithm that achieves $O(T^{\frac{d}{d+2}} d^{\frac{d}{d+2}} +$

$d\sqrt{T \log T}$) regret. This regret rate was proved optimal with a matching lower bound, and was validated through simulations. We also implemented our algorithm on a real-world healthcare Q&A dataset to make decisions on generating new answers v.s. applying an FAQ. Our results open up new avenues for optimizing creation decisions in online learning, with potential extensions to broader loss models and flexible creation costs.

## ETHICS STATEMENT

This research is in accordance with the ICLR Code of Ethics. Our work does not involve human subjects. We use one public dataset under MIT license and one private dataset with proper authorization from the data owner. The proposed methodology is designed to benefit users through improved healthcare Q&A systems, with no identified potential for harm. We have no conflicts of interest to report, and all experimental procedures comply with relevant data usage agreements and licensing terms.

## REPRODUCIBILITY STATEMENT

We have made comprehensive efforts to ensure the reproducibility of our work. All experimental details, including dataset specifications, preprocessing and initialization, evaluation metrics, and statistical significance, are provided in the main text and appendix. Complete algorithmic descriptions and implementation details are included throughout the paper. All theoretical results are accompanied by full proofs in the appendix, with clearly stated assumptions. We commit to releasing the complete source code upon acceptance of this paper. The public dataset used in our experiments is available under MIT license, and we provide detailed descriptions of all data processing steps and experimental procedures to enable replication of our results.

### ACKNOWLEDGMENTS

This work is partially supported by the Engler Family Foundation and the NSF AI Institute for Societal Decision Making (NSF AI-SDM, Award No. 2229881). We appreciate the input from `Nivi.Inc` for providing the maternal health dataset.

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

# Appendix

**Statement of LLM Usage.** We used large language models to assist with (1) text generation, composition, and language polishing, (2) drafting and revising Python code for algorithmic implementations, and (3) literature search in broader fields. All scientific ideas, methodological choices, theoretical analysis and experimental design were conceived, implemented, and validated by the authors, and we have reviewed and verified all LLM-assisted content.

## A  MORE DISCUSSIONS

In this section, we further discuss a few fields and topics of research that are related to our problem modeling, motivation, methodology and implementation.

### A.1  MORE RELATED WORKS

**Online Learning with Resource Constraints.** Another line of related research studies resource-limited bandits, such as "bandits with knapsack (BwK)" (Badanidiyuru et al., 2013) and its versions (Agrawal & Devanur, 2016; Immorlica et al., 2019; Liu et al., 2022). In these scenarios, each arm-pulling consumes some portion of a finite resource (e.g., budget, time, or capacity), and the algorithm aims to optimize the cumulative reward before resources run out. However, these approaches cannot be directly applied to our problem because of a key difference in resource consumption patterns. In BwK, resource consumption only affects the current period's decision-making. In contrast, our setting involves a one-time cost for creating new arms that provides benefits across all future periods through expansion of the action space.

**Digital Healthcare and Clinical Decision Support**  Digital healthcare and clinical decision support systems (CDSS) represent a rapidly evolving field where AI-powered systems must continuously balance the utilization of established medical knowledge with the creation of novel, patient-specific treatment protocols. Foundational work by Rajpurkar et al. (2022) on diagnostic AI systems and the comprehensive framework established by Moor et al. (2023) demonstrate how modern medical AI systems expand beyond narrow, single-task applications to flexible models capable of diverse medical reasoning tasks. Reinforcement learning approaches in critical care, particularly the systematic review by Liu et al. (2017) covering 21 RL applications in intensive care units, illustrate how these systems extend from discrete medication dosing decisions to continuous, multi-dimensional treatment optimization spaces. The create-to-reuse paradigm is particularly evident in precision medicine applications, where systems invest computational resources in developing personalized treatment protocols that can subsequently be applied to patients with similar phenotypic characteristics, effectively creating reusable clinical knowledge that scales across patient populations while maintaining individualized care quality.

**Recommendation Systems and Personalization**  Recommendation systems research has evolved from static collaborative filtering approaches to sophisticated frameworks that dynamically balance the exploitation of existing user preferences with the creation of new personalized recommendation strategies. Neural Collaborative Filtering by He et al. (2017) and the Wide & Deep Learning framework by Cheng et al. (2016) established the foundation for deep learning approaches that can capture complex user-item interactions beyond traditional matrix factorization methods. Meta-learning approaches, particularly by Lee et al. (2019) demonstrate how recommendation systems can treat each user as a distinct learning task, creating personalized model parameters that generalize across different applications and contexts. It is worth mentioning that the multi-armed bandit approaches in recommendation systems (Li et al., 2010) naturally embody the exploration-exploitation-creation tradeoffs by continuously balancing known user preferences with the discovery of new content types and recommendation strategies. Our create-to-reuse framework directly parallels these systems' core functionality: recommendation systems routinely invest computational resources in creating personalized embeddings, meta-learned initialization parameters, and graph neural network representations that serve as reusable templates for rapid adaptation to new users, items, and interaction modalities, while continuously expanding their action spaces through dynamic catalog growth and emerging user behavior patterns.

**Active Learning** Active learning frameworks fundamentally embody the exploration-exploitation-creation paradigm by allowing algorithms to strategically choose their training data, thereby naturally connecting to sequential decision-making with expanding action spaces. Settles (2009) established the theoretical foundations for query selection strategies, while membership query synthesis approaches (Angluin, 1988) demonstrated how active learners can create entirely new query types rather than merely selecting from existing unlabeled data pools. Query-by-Committee methods (Seung et al., 1992) and extended through frameworks like QUIRE by Huang et al. (2010) show how multiple learning strategies can be combined to create adaptive query selection policies that balance informativeness and representativeness. Closer work on meta-active learning and the "Growing Action Spaces" framework by Farquhar et al. (2020) directly address expanding action spaces through curriculum learning approaches that progressively grow query complexity. The create-to-reuse framework maps directly onto active learning's core mechanisms: systems invest computational effort in synthesizing new query types, developing committee-based strategies, and learning meta-policies for query selection, creating reusable query generation mechanisms and adaptive selection strategies that can be applied across different datasets, domains, and learning tasks, while continuously expanding their query capabilities as they encounter new data distributions and learning scenarios.

**Inventory Management** Inventory management and supply chain systems represent a mature operations research domain where organizations continuously face fundamental tradeoffs between optimizing existing supply chain capabilities and investing in new suppliers, products, or distribution channels. Bellman (1958) established the mathematical foundations of inventory theory, while dynamic capacity expansion models (Mieghem & Rudi, 2002) demonstrate how firms balance existing capacity utilization with flexible resource investments that create new operational capabilities. The create-to-reuse framework aligns naturally with supply chain decision-making: organizations invest upfront in new suppliers, products, or distribution capabilities that become reusable assets for future deployment across different demand scenarios.

### A.2 POTENTIAL EXTENSION AND GENERALIZATION

**Dynamic and Context-Dependent Creation Costs.** Our current framework assumes a fixed creation cost $c$ across all time steps and contexts. A natural extension would allow time-varying costs $c_t$ or context-dependent costs $c(x_t)$ that reflect realistic scenarios where creation difficulty varies with problem complexity or resource availability. This generalization would better capture applications like drug discovery, where synthesis costs depend on molecular complexity, or content generation, where review costs vary with topic sensitivity. However, this extension introduces significant algorithmic challenges, as evidenced by the substantially worse competitive ratios in variant-cost online facility location problems, where even achieving constant competitive ratios becomes impossible under adversarial sequences.

**Non-Parametric and Neural Function Approximation.** While our theoretical analysis focuses on parametric quadratic loss functions $d(x, f)$, our empirical experiments demonstrate promising results when replacing the distance function with neural networks and using LLM-as-a-judge for feedback evaluation. Extending the theoretical guarantees to broader function classes, particularly neural networks or kernel methods, would significantly broaden the applicability of our framework. The key challenge lies in controlling the complexity of the function class while maintaining meaningful regret bounds, potentially requiring techniques from neural tangent kernels (NTK) or Bayesian optimization (BO) to handle the high-dimensional hypothesis space.

**Scalability and Long-Term Operation.** The algorithm presented in the main paper requires, at each time step t, a search over the entire library of existing contexts $S_t$ to find the one that minimizes the Lower Confidence Bound (LCB). A critical consideration for practical deployment in long-running systems is the scalability of this step as the library size $|S_t|$ grows.

1. To empirically characterize the growth of the library, we ran simulations and recorded $|S_t|$ with respect to $t$. The results are displayed in Figure 3. We observe that in high-dimensional embedding spaces, $|S_t|$ tends to grow almost linearly with $t$. This is because the expected number of generated actions are $O(T^{\frac{d}{d+2}}) \approx O(T^1)$ as $d$ being very large. Intuitively, a new

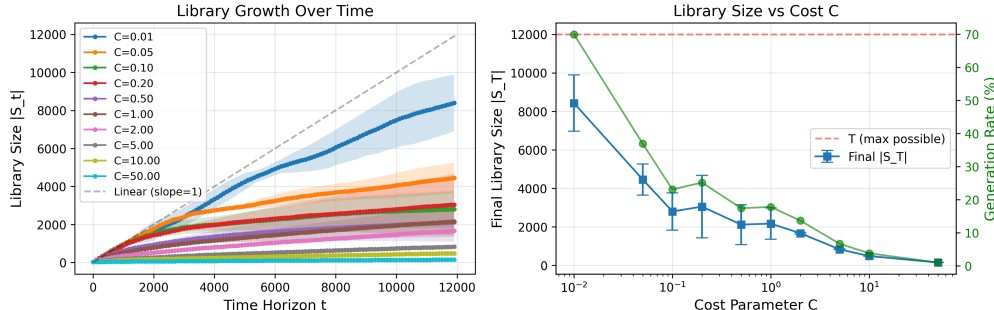

Figure 3: Library growth dynamics under varying creation costs $c$. (Left) The action library size $|S_t|$ grows slightly sub-linearly with time $t$ (below the dashed slope-1 line), with growth rate decreasing as $t$ increases. (Right) The creation cost parameter $c$ critically controls the creation-reuse tradeoff: higher $c$ leads to substantially fewer library elements (blue) and lower generation rates (green), as the algorithm favors reusing existing actions despite greater mismatch loss. Error bars show standard deviation across runs.

    context in high dimensions $x_t$ is frequently far from all existing library items $f \in S_t$, making the creation of a new item more likely than reuse.

2. Our experiments show that the creation cost $c$ is the most significant hyperparameter for controlling library size. As seen in Figure 3, higher values of c substantially slow the growth of $|S_t|$, as the algorithm is more heavily penalized for creating new items and is thus incentivized to tolerate a higher modeling loss from reusing an existing item. This provides a direct and practical lever for system designers to manage the trade-off between computational cost and the richness of the action library.

The potential for linear growth in $|S_t|$ necessitates strategies to manage computational complexity. While the naive $O(|S_t|)$ search is feasible for moderate $T$, large-scale systems would benefit from the following practical optimizations.

1. **Approximate Nearest-Neighbor (ANN) Search:** The core of the selection step is to find $\arg\min_{f \in S_t} LCB(x_t, f)$. The LCB is a function of the learned distance $d(x_t, f) = (x_t - f)^\top W(x_t - f)$. This implies that the minimizing $f$ is very likely to be one of the nearest neighbors to $x_t$ under this learned distance norm defined by $\hat{W}$ (an estimate of $W$). Instead of an exhaustive linear scan, we can use established ANN libraries to retrieve the $k$ nearest neighbors of $x_t$ from $S_t$ in sublinear time (often $O(log|S_t|)$). The LCB computation can then be restricted to this small candidate set. This optimization becomes increasingly reliable as the learned matrix $\hat{W}$ converges to a good representation of the underlying task structure.

2. **Library De-activation and Caching:** The theoretical result that an optimal covering of the context space has a sublinear size of $O(T^{d/(d+2)})$ suggests that if $|S_t|$ grows linearly, the library must contain a high degree of redundancy. This insight motivates a "de-activation" or caching strategy.

   (i) We can maintain an "active set" of library items, which is a subset of the full library $S_t$. At each step, the LCB computation is only performed over this active set.

   (ii) The active set could be managed using heuristics such as Least Recently Used (LRU), where items that have not been selected for a long time are moved to an inactive state. More sophisticated methods could periodically re-compute a core-set or a sparse covering of the full library $S_t$ to form the active set, ensuring that the entire space of created items is still represented. This approach contains the per-step computational cost while preserving the full library for long-term knowledge retention, aligning the practical implementation with the theoretical sublinear nature of the optimal solution.

**Additional Baseline: LinLCB.** We implemented a LinLCB baseline (a version of LinUCB (Chu et al., 2011) in terms of our loss-minimization setting) to select the best existing action, paired with

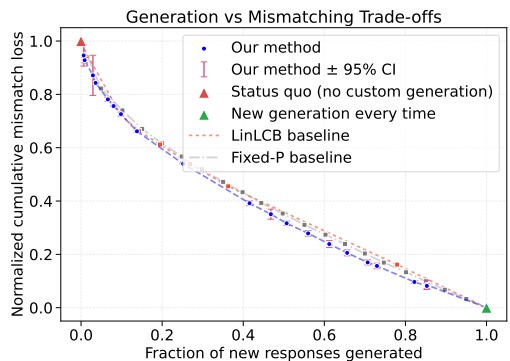

Figure 4: Comparison of creation-reuse strategies on the Medical Q&A dataset. The LinLCB baseline (using LCB both for action selection and creation threshold) performs worse than our doubly-optimistic method and frequently falls above (worse than) the Fixed-P baseline. This degradation stems from a coupling effect: conservative LCB-based creation decisions suppress early library growth, which reduces exploration capacity and compounds into a negative feedback loop of insufficient learning and higher mismatch loss.

a deterministic creation rule that creates iff $\min_f LCB_t(x_t, f) > c'$, and we swept the threshold $c'$ (which is different from the creation cost $c$) to plot the generation-mismatch tradeoff curve as we did in Section 6.2. This numerical experiment is conducted on the Medical Q&A dataset, and we compare the performance of LinLCB against our method (doubly-optimistic) and the Fixed-$P$ baseline. The results are shown in Figure 4.

Empirically, this LinLCB baseline performs worse than our method, and it even goes above (worse than) the Fixed-P baseline frequently. The main reason is a coupling effect: using an LCB both to select among existing actions and to trigger creation makes creation excessively conservative in early rounds (LCBs are low under high uncertainty). This suppresses the addition of new actions, which in turn limits future exploration capacity and compounds the problem—leading to fewer candidates and higher mismatch later. It end up getting a negative feedback loop:

(i) Optimism in LinLCB lowers the best available LCB.

(ii) This suppresses the creation of new, necessary library items.

(iii) The library $S_t$ remains small and insufficient, leading to poor learning and high modeling error in the long run.

In contrast, the "Fixed-P" baseline, while naive, ensures a constant and stable rate of library growth, preventing this collapse into a state of insufficient learning. This result highlights the critical difference between "exploration" versus "creation" that might be misunderstood intuitively. To this extent, our "doubly-optimistic" framework does not just optimistically evaluate existing actions; it optimistically evaluates the choice to create a new action by treating $c$ as its potential value.

## B  PROOF DETAILS

Here we extend the proof sketch of Theorem 5.4 provided in Section 5. According to the roadmap depicted, to validate Theorem 5.4, we only need to prove the following Lemmas B.1, B.3 and B.8. We first propose the lemma that bounds $OPT_o$.

**Lemma B.1** ($OPT_o$ upper bound). *We have $OPT_o = O(T^{\frac{d}{d+2}} d^{\frac{d}{d+2}})$.*

*Proof sketch.* We propose a context set (library) $\tilde{S}$ such that $c \cdot |\tilde{S}| + \sum_{t=1}^{T} \min_{f \in \tilde{S}} d(x_t, f) = O(T^{\frac{d}{d+2}} d^{\frac{d}{d+2}})$. Specifically, we let $\tilde{S} := \{[N_1, N_2, \ldots, N_d]^\top | N_i \in [\frac{1}{\Delta}], i = 1, 2, \ldots, d\}$ as a $\Delta$-covering set over the context space of $[0, 1]^d$. On the one hand, the cumulative mismatch loss due to

discretization of the context space is $O(T \cdot \Delta^2 d)$. On the other hand, the total cost of adding new contexts to the set is $O((\frac{1}{\Delta})^d)$. Let $\Delta = T^{-\frac{1}{d+2}} d^{-\frac{1}{d+2}}$ and the total loss is $O(T^{\frac{d}{d+2}} d^{\frac{d}{d+2}})$. Please kindly find a detailed proof in Appendix B.1. $\qquad\square$

Before getting into the main lemma that upper bounds $ALG$, we present another lemma showing the concentration of $\bar{d}_t(x_t, f)$ within $\Delta_t(x_t, f)$.

**Lemma B.2** ($\Delta_t$ as estimation error). *The estimation error of $|\bar{d}_t(x_t, f) - d(x_t, f)|$ is upper bounded by $\Delta_t(x_t, f)$ with high probability. As a consequence, we have $d(x_t, f) - 2\Delta_t(x_t, f) \leq \check{d}_t(x_t, f) \leq d(x_t, f) \leq \hat{d}_t(x_t, f) \leq d(x_t, f) + 2\Delta_t(x_t, f)$.*

The proof of Lemma B.2 is deferred to Appendix B.2. In the following, we state the lemma that upper bounds the algorithmic loss by a constant competitive ratio over $OPT_o$ adding estimation errors. According to

**Lemma B.3** (Constant competitive ratio). *We have $ALG \leq 60 OPT_o + 54 \sum_{t=1}^{T} \Delta_t(x_t, f_t)$.*

*Proof.* Before starting the proof, we emphasize that all operations we make in this proof are made in the *context* space. As we frequently mention in this paper, the actions are only accessible through the oracle $\mathcal{A}(x)$ for some context $x$. Therefore, the context library $S_t$ is sometimes referred as a "set" without causing misunderstandings.

First of all, we note that the following two $\{x_t\}_{t=1}^{T}$ series have identical joint distributions:

(a) Sample a sequence of $x_1, x_2, \ldots, x_T$ independently from an identical distribution $\mathbb{D}_X$. (iid)

(b) Sample a set of $Z := \{z_1, z_2, \ldots, z_T\}$ independently from the identical distribution $\mathbb{D}_X$, and then sample $\{x_t\}_{t=1}^{T}$ as a uniformly random permutation of $Z$, i.e. $\{x_t\}_{t=1}^{T} \sim U(\sigma(Z))$. Here $\sigma(Z)$ denotes the permutation set of $Z$. (iid + permutation)

Given this property, we assume that $\exists Z = \{z_1, z_2, \ldots, z_T\}, z_t \overset{\text{i.i.d.}}{\sim} \mathbb{D}_X, \{x_t\}_{t=1}^{T} \sim U(\sigma(Z))$. In the following, we will keep using the notations of $\{x_t\}_{t=1}^{T}$ and $Z$ accordingly.

Consider the optimal offline solution $S^*$ such that

$$
\begin{aligned}
OPT_o &= c \cdot |S^*(x_1, x_2, \ldots, x_T)| + \sum_{t=1}^{T} \min_{f \in S^*} d(x_t, f) \\
&= c \cdot |S^*(z_1, z_2, \ldots, z_T)| + \sum_{t=1}^{T} \min_{f \in S^*} d(x_t, f).
\end{aligned}
\tag{9}
$$

Here we denote $S^*(x_1, x_2, \ldots, x_T)$ and $S^*(z_1, z_2, \ldots, z_T)$ differently to show that the offline solution is not dependent on the permutation, with slight abuse of notation. Denote $S^* =: \{c_1^*, c_2^*, \ldots, c_K^*\}$. For each $c_i^*, i = 1, 2, \ldots, K$, denote a subset of $\{x_t\}$ as $C_i^*$ such that $\min_{f \in S^*} d(x_t, f) = d(x_t, c_i^*), \forall x_t \in C_i^*$. In other words, $C_i^*$ consists of all $x_t$'s that are assigned to $c_i^*$ in the optimal solution $S^*$. Denote $A_i^* := \sum_{t: x_t \in C_i^*} d(x_t, c_i^*)$ as the total optimal cost associated with $c_i^*$, and $a_i^* := \frac{A_i^*}{|C_i^*|}$ as the average cost in $C_i^*$.

Now, we define $C_i^g$ and $C_i^b$ as separated GOOD and BAD subsets of $C_i^*$, respectively, such that

$$
\begin{aligned}
C_i^g &\subset C_i^*, \ C_i^b \subset C_i^*, \ |C_i^g| = |C_i^b| = \frac{|C_i^*|}{2} \\
d(x_g, c_i^*) &\leq d(x_b, c_i^*), \forall x_g \in C_i^g, x_b \in C_i^b.
\end{aligned}
\tag{10}
$$

In other words, $C_i^g$ and $C_i^b$ represent the nearest half and the farthest half of $x_t$'s in the set $C_i^*$, in terms of distance to $c_i^*$. Note that the sets $C_i^g$ and $C_i^b$ are determined by $Z$ and not relevant to the permutation. Therefore, once $Z$ is realized, the random sequence $\{x_t\}_{t=1}^{T}$ does not affect $C_i^g$ and $C_i^b$.

Given these notations, we present and prove the following two lemmas: a Lemma B.4 bounding the *total* loss of GOOD $x_t$'s, and a Lemma B.6 bounding the *individual* loss of each BAD $x_t$'s.

**Lemma B.4.** *The total loss caused by all $x_t \in C_i^g$ is upper bounded as*

$$\sum_{t:x_t \in C_i^g} \mathbb{E}[l_t | \{x_t\}_{t=1}^T] \leq 3c + 4A_i^* + 4 \sum_{x_t \in C_i^g} d(x_t, c_i^*) + 6 \sum_{t=1}^T \Delta_t(x_t, f_t). \tag{11}$$

*Proof of Lemma B.4.* Denote the context set (library) sequence as $\{S_t\}_{t=1}^T$. Also, denote $\Delta_t := \Delta_t(x_t, f_t)$ and $d_t^* := d(x_t, c_i^*)$ for simplicity. In fact, any $x_t \in C_i^g$ falls in one of the following two cases:

(I) When $\exists\, e_i \in S_T$ such that $d(e_i, c_i^*) \leq 2a_i^*$, we further categorize $x_t$ into three sub-cases:

**I.(a).** At time $t$, we select context $e_i$ and deploy $a_t = \mathcal{A}(e_i)$ (i.e., $x_t$ is matched to context $e_i$). We have

$$d(x_t, e_i) \leq 2(d(x_t, c_i^*) + d(c_i^*, e_i)) \leq 2(d_t^* + 2a_i^*). \tag{12}$$

The first inequality is due to

$$d(a, b) + d(b, c) \geq \frac{1}{2} d(a, c), \forall a, b, c \in \mathbb{R}^{d^2}. \tag{13}$$

as a quadratic form. Hence

$$\mathbb{E}[l_t | \{x_t\}_{t=1}^T] \leq 2d(x_t, e_i) + 2\Delta(x_t, f_t) \leq 4(d_t^* + 2a_i^*) + 2\Delta_t. \tag{14}$$

**I.(b).** At time $t$, $e_i \in S_t$ but $a_t \neq \mathcal{A}(e_i)$, i.e. $x_t$ is matched to some other context $f_t$ even with the existence of $e_i$. Now we have

$$d(x_t, f_t) - 2\Delta_t \leq \check{d}_t(x_t, f_t) \leq \check{d}_t(x_t, e_i) \leq d(x_t, e_i). \tag{15}$$

The second inequality comes from the arg-minimum definition of $f_t$, and the first and third inequalities is from Lemma B.2. Therefore, we have

$$d(x_t, f_t) \leq d_t(x_t, e_i) + 2\Delta_t \leq 2(d(x_t, c_i^*) + d(c_i^*, e_i)) + 2\Delta_t \leq 2(d_t^* + 2a_i^*) + 2\Delta_t \tag{16}$$

Hence we have

$$\mathbb{E}[l_t | \{x_t\}_{t=1}^T] \leq 2d(x_t, e_i) + 2\Delta_t \leq 4(d_t^* + 2a_i^*) + 6\Delta_t. \tag{17}$$

**I.(c).** $e_i \notin S_t$ at time $t$, i.e. $x_t$ is matched to some $f_t$ before any close-enough context $e_i$ being added. In this case, we propose the following lemma that provides an *overall* loss bound for any group of $\{x_t\}$'s, on which no new actions have been created.

**Lemma B.5** (Constant loss bound before a new action being generated). *Denote $Q := \{x_{t_i},\ i = 1, 2, \ldots, n | 1 \leq t_1 \leq \ldots \leq t_n \leq T\}$ as a subsequence of $\{x_t\}_{t=1}^T$. Also, denote $t_k$ as the first time in $Q$ such that a new action is generated, i.e. $a_{t_k} = \mathcal{A}(x_{t_k})$ and $a_{t_i} \neq \mathcal{A}(x_{t_i}), i \leq k - 1$. We have*

$$\mathbb{E}[\sum_{i=1}^{k-1} l_{t_i} | \{x_t\}_{t=1}^T] \leq c. \tag{18}$$

We defer the proof of Lemma B.5 to Appendix B.3, where we will prove a generalized claim. According to Lemma B.5, the total expected loss for all $x_t$ in this case can be bounded by $c$.

(II) When $\forall\, e \in S_T$ satisfies $d(e, c_i^*) > 2a_i^*$, we know that no new action are generated at time $t, \forall t:\ x_t \in C_i^g$. Then we again apply Lemma B.5 and upper bound the expected total loss by $c$.

Combining Case I (a,b,c) and Case II, along with a separate cost $c$ of adding $e_i$, we have an upper bound on the expected total loss for all $t : x_t \in C_i^g$ as follows:

$$\mathbb{E}[\sum_{t:x_t \in C_i^g} l_t] \leq 4 \sum_{t:x_t \in C_i^g} d_t^* + 8 \sum_{t:x_t \in C_i^g} a_i^* + 6 \sum_{t:x_t \in C_i^g} \Delta_t + 3c$$

$$= 4 \sum_{t:x_t \in C_i^g} d_t^* + 4A_i^* + 6 \sum_{t:x_t \in C_i^g} + 3c. \tag{19}$$

Here the last line comes from $|C_i^g| = \frac{|C_i^*|}{2}$. This proves Lemma B.4. $\square$

The previous lemma bounds the *total* loss of GOOD $x_t$'s, while the following lemma will bound the *individual* loss of BAD $x_t$'s".

**Lemma B.6.** *For each individual $x_t \in C_i^b$, the expected loss is upper bounded as*

$$\mathbb{E}[l_t | Z] \leq 4d(x_t, c_i^*) + 4\Delta_t(x_t, f_t) + \frac{2}{|C_i^*|} \cdot (c + 8 \sum_{s:x_s \in C_i^g} \mathbb{E}[l_s | Z] + 8 \sum_{s:x_s \in C_i^g} d(x_s, c_i^*)). \tag{20}$$

*Proof sketch of Lemma B.6.* Intuitively, *later*-arrived $x_t$'s should be facing a *better* situation as there are more action candidates. Therefore, for any $x_t \in C_i^b$, if there exists a good point $x_g$ that emerges before the occurrence of $x_t$, we can upper bound $\mathbb{E}[l_t]$ with $\mathbb{E}[l_g]$ adding $d(x_t, c_i^*)$. This is because we can at least match $x_t$ to the existing in-library context that $x_g$ was matched to. Denote $f_g$ as the existing context whose custom action $x_g$ was assigned to. According to the "triangular inequality" shown as Equation (13) (up to constant coefficient), we have: $E[l_t] \leq O(d(x_t, f_g)) \leq O(d(x_t, c_i^\star) + d(c_i^\star, f_g)) = O(d(x_t, c_i^\star) + d(c_i^\star, x_g) + d(x_g, f_g)) = O(d(x_t, c_i^\star) + d(x_g, c_i^\star) + E[l_g])$. If there is no such a $x_g$ (with very small probability), then we upper bound the expected loss by $c$. For a detailed proof of Lemma B.6, please kindly refer to Appendix B.4. $\square$

Combining Lemma B.4 and Lemma B.6 above, we have

$$\mathbb{E}[\sum_{t:x_t \in C_i^*} l_t]$$

$$= \mathbb{E}[\mathbb{E}[\sum_{t:x_t \in C_i^*} l_t | Z]]$$

$$= \mathbb{E}[\mathbb{E}[\sum_{s:x_s \in C_i^g} l_s | Z] + \mathbb{E}[\sum_{r:x_r \in C_i^b} l_r | Z]]$$

$$= \mathbb{E}[\mathbb{E}[\sum_{t:x_t \in C_i^*} l_t | \{x_t\}_{t=1}^T] + \mathbb{E}[\sum_{r:x_r \in C_i^b} l_r | Z]]$$

$$\leq \mathbb{E}[3c + 4A_i^* + 4 \sum_{s:x_s \in C_i^g} d(x_s, c_i^*) + 6 \sum_{s:x_s \in C_i^g} \Delta_s(x_s, f_s)] \tag{21}$$

$$+ \mathbb{E}[4 \sum_{r:x_r \in C_i^b} d(x_r, c_i^*) + 4 \sum_{r:x_r \in C_i^b} \Delta_r(x_r, f_r)$$

$$+ \frac{|C_i^*|}{2} \cdot \frac{2}{|C_i^*|} \cdot (c + 8 \sum_{s:x_s \in C_i^g} \mathbb{E}[l_s] + 8 \sum_{s:x_s \in C_i^g} d(x_s, c_i^*))]$$

$$\leq \mathbb{E}[28c + 40A_i^* + 40 \sum_{s:x_s \in C_i^g} d(x_s, c_i^*) + 54 \sum_{t:x_t \in C_i^*} \Delta_t(x_t, f_t)]$$

$$\leq \mathbb{E}[28c + 60A_i^* + 54 \sum_{t:x_t \in C_i^*} \Delta_t(x_t, f_t)].$$

Here the last inequality is because $\sum_{s:x_s \in C_i^g} d(x_s, c_i^*) \leq \frac{\sum_{s:x_s \in C_i^g} + \sum_{r:x_r \in C_i^b}}{2} = \frac{A_i^2}{2}$. On the other hand, the sum of losses in $OPT_o$ that are associated to $c_i^*$ equals $c + A_i^*$. Therefore, we have $ALG \leq 60OPT_o + 54 \sum_{t=1}^T \Delta_t(x_t, f_t)$. This ends the proof of Lemma B.3. $\square$

*Remark* B.7. The reason for us to divide $\{x_t\}$'s into GOOD and BAD subsets is twofold.

(1) We can upper-bound the *total* loss of all GOOD points, mainly because we have Lemma B.5 such that the Case I(c) and Case II hold. Lemma B.5 states that for any group of $\{x_t\}$'s, the expected cost before a new action being created (i.e. before a new context is added to the library) among them is no more than $c$. Therefore, if there does not exist an $e_i$ close enough to $c_i^*$, we know that no new actions have been created among GOOD $\{x_t\}$'s (since any GOOD $x_t$ satisfies $d(x_t, c_i^*) \le 2a_i^*$ and therefore is a qualified candidate $e_i$ once being added to the existing context library). However, this does not hold for BAD points, as they may still trigger new action generations although their contexts are faraway from $c_i^*$.

(2) We can only upper-bound the *individual* loss of each BAD $x_t$ due to the reason in (1) above. The individual upper bound for a BAD point is applicable for a GOOD point, but this would introduce a linear dependence on $T \cdot c$ in the overall loss instead of a constant ratio.

Now we propose the lemma where we upper bound the cumulative estimation error.

**Lemma B.8** (Linear regression excess risk). *The cumulative absolute error of online linear regression with least-square estimator satisfies* $\sum_{t=1}^{T} \Delta_t = O(\sqrt{d^2 T \log T})$.

We defer the proof of Lemma B.8 to Appendix B.5 as a standard result from linear regression. According to what we stated earlier, this completes the proof of Theorem 5.4.

In the following subsections, we present the proof details of lemmas proposed above.

## B.1 PROOF OF LEMMA B.1

*Proof.* Let $\tilde{S} = \{[N_1, N_2, \ldots, N_d]^\top | N_i \in [\frac{1}{\Delta}], i = 1, 2, \ldots, d\}$. On the one hand, for any context $x = [x_1, x_2, \ldots, x_d]^\top \in \mathbb{R}^d, \|x\|_2 \le 1$, consider $f_x := [\lfloor \frac{x_1}{\Delta} \rfloor \cdot \Delta, \lfloor \frac{x_2}{\Delta} \rfloor \cdot \Delta, \ldots, \lfloor \frac{x_d}{\Delta} \rfloor \cdot \Delta]$. Due to the definition of $\tilde{S}$, we know that $f_x \in \tilde{S}$. Also we have $d(x, f_x) = \|x - f_x\|_W^2 < \|[\Delta, \Delta, \ldots, \Delta]^\top\|_W^2 \le \lambda_{\max}(W) \cdot \Delta^2 d$. On the other hand, we have $|\tilde{S}| = (\frac{1}{\Delta})^d$. Denote $S^*$ as the solution to $OPT_o$ (as defined in Equation (9)), we have

$$
\begin{aligned}
OPT_o =& c \cdot |S^*| + \sum_{t=1}^{T} \min_{f \in S^*} d(x_t, f) \\
\le& c \cdot |\tilde{S}| + \sum_{t=1}^{T} \min_{f \in \tilde{S}} d(x_t, f) \\
\le& c \cdot (\frac{1}{\Delta})^d + \sum_{t=1}^{T} d(x_t, f_{x_t}) \\
\le& c(\frac{1}{\Delta})^d + T \cdot \lambda_{\max}(W) \cdot \Delta^2 d \\
=& O(\frac{1}{\Delta^d} + T\Delta^2 d),
\end{aligned}
\tag{22}
$$

and we let $\Delta = T^{-\frac{1}{d+2}} d^{-\frac{1}{d+2}}$ to make the RHS $= O(T^{\frac{d}{d+2}} d^{\frac{d}{d+2}})$. This proves the lemma. $\square$

## B.2 PROOF OF LEMMA B.2

*Proof.* Here we prove a more general result on ridge regression:

**Lemma B.9.** *Let* $x_1, x_2, \ldots, x_n \in \mathbb{R}^d$ *are d-dimension vectors, and* $y_i := x_i^\top \theta^* + N_t$, *where* $\theta^* \in \mathbb{R}^d$ *is a fixed unknown vector such that* $\|\theta^*\|_2 \le 1$ ,*and* $N_t$ *is a* martingale difference sequence *subject to $\sigma$-subGaussian distributions. Denote* $X = [x_1, x_2, \ldots, x_n]^\top \in \mathbb{R}^{n \times d}$ *and* $Y = [y_1, y_2, \ldots, y_n]^\top \in \mathbb{R}^n$. *Let the ridge regression estimator*

$$
\hat{\theta} := (X^\top X + I_d)^{-1} X^\top Y
$$

where $I_d$ is the $d \times d$ identity matrix. Then with probability $\Pr \geq 1 - \delta$, we have

$$|x^\top(\theta^* - \hat{\theta})| \leq O\left((1 + \sqrt{\log(\frac{2}{\delta})})\sqrt{x^\top(X^\top X + I_d)^{-1}x}\right) \tag{23}$$

holds for any $\delta > 0$ and $x \in \mathbb{R}^d$.

*Proof of Lemma B.9.* Denote $N := [N_1, N_2, \ldots, N_n]^\top \in \mathbb{R}^n$ as the vector of noises in the labels. Then we have

$$\hat{\theta} = (X^\top X + I_d)^{-1}X^\top X\theta^* + (X^\top X + I_d)^{-1}X^\top N. \tag{24}$$

Therefore, the difference between $\theta^*$ and $\hat{\theta}$ can be characterized as

$$\begin{aligned}
\theta^* - \hat{\theta} &= \theta^* - (X^\top X + I_d)^{-1}X^\top X\theta^* - (X^\top X + I_d)^{-1}X^\top N \\
&= (X^\top X + I_d)^{-1}(X^\top X + I_d)\theta^* - (X^\top X + I_d)^{-1}X^\top X\theta^* - (X^\top X + I_d)^{-1}X^\top N \\
&= (X^\top X + I_d)^{-1}\theta^* - (X^\top X + I_d)^{-1}X^\top N \\
&= (X^\top X + I_d)^{-1}(\theta^* - X^\top N).
\end{aligned} \tag{25}$$

As a result, we have

$$\begin{aligned}
|x^\top(\theta^* - \hat{\theta})| &= |x^\top(X^\top X + I_d)^{-1}(\theta^* - X^\top N)| \\
&\leq |x^\top(X^\top X + I_d)^{-1}\theta^*| + |x^\top(X^\top X + I_d)^{-1}X^\top N|.
\end{aligned} \tag{26}$$

For the simplicity of notation, denote $A := (X^\top X + I_d)^{-1}$, then we have $|x^\top(\theta^* - \hat{\theta})| \leq \|x^\top A\theta^*\|_2 + \|x^\top AX^\top N\|$. On the one hand, for the first term we have

$$\begin{aligned}
|x^\top A\theta^*| &\leq \|A^\top x\|_2 \cdot \|\theta^*\|_2 \\
&\leq \sqrt{x^\top AA^\top x} \cdot 1 \\
&\leq \sqrt{x^\top Ax}.
\end{aligned} \tag{27}$$

The second line is because $\|\theta^*\| \leq 1$, and the last inequality is because $A = A^\top = (X^\top X + I_d)^{-1} \prec I_d$.

On the other hand, for the second term, recall that we set $A := (X^\top X + I_d)^{-1}$ and $\theta^* - \hat{\theta} = A(\theta^* - X^\top N)$. We consider the random variable $x^\top AX^\top N = \sum_{t=1}^n \alpha_t N_t$, where the deterministic coefficients $\alpha_t := (x^\top AX^\top)_t, \quad t = 1, \ldots, n$.

Notice that $\{N_t\}$ is a martingale difference sequence with subGaussian tails. According to Jin et al. (2019, Proposition 7), which is a subGaussian version of Azuma–Hoeffding's Inequality, let $d = 1$ and there exists a constant $C_J$ such that

$$\left|\sum_{t=1}^n \alpha_t N_t\right| \leq C_J \cdot \sqrt{\sum_{t=1}^n \alpha_t^2 \log\frac{2}{\delta}}. \tag{28}$$

with probability $\Pr \geq 1 - \delta$. Here $\|\alpha\|_2^2 = \sum_{t=1}^n \alpha_t^2 = x^\top AX^\top X A x \leq x^\top Ax$ because $X^\top X \preceq X^\top X + I_d$.

Therefore, with probability at least $1 - \delta$,

$$|x^\top AX^\top N| \leq C_J\sqrt{x^\top Ax \log\left(\frac{2}{\delta}\right)}. \tag{29}$$

Returning to $|x^\top(\theta^* - \hat{\theta})| \leq |x^\top A\theta^*| + |x^\top AX^\top N|$, we already established (using $\|\theta^*\|_2 \leq 1$) that $|x^\top A\theta^*| \leq \sqrt{x^\top Ax}$. Combining this with Equation (29) as the martingale tail bound, we get $|x^\top(\theta^* - \hat{\theta})| \leq \sqrt{x^\top Ax} + C_J \cdot \sqrt{x^\top Ax \log\left(\frac{2}{\delta}\right)} = \left(1 + C_J \cdot \sqrt{\log\left(\frac{2}{\delta}\right)}\right)\sqrt{x^\top(X^\top X + I_d)^{-1}x}$.

This ends the proof of Lemma B.9.

$\square$

Now let us go back to the proof of Lemma B.2. We apply this lemma for $6T$ times: in the proof of Lemma B.4 as Case I(a), I(b), I(c) (or Lemma 5.6) and II, and in the proof of Lemma B.6 as Case I and Case II, in each of which we adopt this concentration bound for each existing context $f \in S_t$, which is at most $T$. Therefore, we let $\delta \leftarrow \frac{1}{6T^2}\delta$ and let $\lambda = 1$, $\alpha = (1 + C_J \cdot \sqrt{\log \frac{12T^2}{\delta}}) \cdot \|W\|_F$. According to Lemma B.9, we prove that $\bar{d}(x_t, f) - \Delta_t(x_t, f) \leq d(x_t, f) \leq \bar{d}(x_t, f) + \Delta_t(x_t, f)$ holds for any $f \in S_t$ and $\forall t = 1, 2, \ldots, T$, with probability $\Pr \geq 1 - \delta$. Therefore, we have proved Lemma B.2.

$\square$

### B.3 PROOF OF LEMMA B.5

*Proof.* Notice that at each time $t_k$, with probability $\Pr = \frac{\hat{d}_{t_k}(x_{t_k}, f_{t_k})}{c}$ we terminate this stochastic process, and with the rest $\Pr = 1 - \frac{\hat{d}_{t_k}(x_{t_k}, f_{t_k})}{c}$ we add $d_{t_k}(x_{t_k}, f_{t_k})$ to our cumulative expected loss. Since $\hat{d}_{t_k}(x_{t_k}, f_{t_k}) \geq d_{t_k}(x_{t_k}, f_{t_k}), \forall k \in [n]$, we may instead prove a generalized version of this lemma.

**Lemma B.10.** *Consider an infinite sequence $\{p_1, p_2, \ldots, p_k, \ldots\}$ where $p_k \in [0, 1]$. The initial sum $S = 0$. At each time $k$, with probability $p_k$ we stop this stochastic process, otherwise we add $p_k$ to the sum $S$. We show that $\mathbb{E}[S] \leq 1$.*

Lemma B.10 is a generalization of Lemma B.5 since we add $d_{t_k}(x_{t_k}, f_{t_k}) \leq \hat{d}_{t_k}(x_{t_k}, f_{t_k})$ at each time $k$ in the latter setting.

Denote a random variable $I_k$ as follows: $I_k = 1$ if the stochastic process has not stopped by the end of time $k$, and $I_k = 0$ otherwise. In the case when $I_k = 1$, we add $p_k$ to the sum $S$. Therefore, we have

$$\mathbb{E}[S] = \sum_{k=1}^{\infty} p_k I_k$$

. Also, we know that the probability that $I_k = 1$ is $\Pr[I_k = 1] = \prod_{i=1}^{k}(1 - p_i)$. As a result, we have

$$\mathbb{E}[S] = \mathbb{E}[\sum_{k=1}^{\infty} p_k \cdot I_k]$$
$$= \sum_{k=1}^{\infty} p_k \prod_{i=1}^{k}(1 - p_i). \tag{30}$$

In the following, we show that $\sum_{k=1}^{\infty} p_k \prod_{i=1}^{k}(1 - p_i) \leq 1$. We first consider $p_k \in (0, 1)$. Denote $Q_0 := 1$ and $Q_k := \Pr[I_k = 1] = \prod_{i=1}^{k}(1 - p_i)$, and we know $Q_k = (1 - p_k)Q_{k-1} \leq Q_{k-1}$. Also, we have $p_k Q_{k-1} = (1 - (1 - p_k))Q_{k-1} = Q_{k-1} - Q_k$.

For the rigorousness of the proof, we first show that $\sum_{k=1}^{\infty} p_k Q_k$ is finite. Denote

$$T_n := \sum_{k=1}^{n} p_k Q_k, \tag{31}$$

and we have

$$T_n \leq \sum_{k=1}^{n} p_k Q_{k-1}$$
$$= \sum_{k=1}^{n} Q_{k-1} - Q_k \tag{32}$$
$$= Q_0 - Q_n$$
$$< Q_0 = 1$$

As $T_n < 1$ and $T_{n+1} \geq T_n, \forall n \geq 1$, we have

$$\lim_{n \to \infty} T_n \leq 1 \tag{33}$$

according to the Monotone Convergence Theorem. Then we slightly generalize the results above from $p_k \in (0,1)$ to $p_k \in [0,1]$, i.e. incorporating 0 and 1. In fact, if $p_k = 0$, then we may skip this $p_k Q_k$ term. Otherwise if $p_k = 1$, consider the first $m$ s.t. $p_m = 1$, and then we still have $E[S] = \sum_{k=1}^{m-1} p_k I_k = T_{m-1} < 1$ and $I_m = I_M = 0$ for any $M \geq m, M \in \mathbb{Z}^+$.

Therefore, we have

$$
\begin{aligned}
\mathbb{E}[S] &= \sum_{k=1}^{\infty} p_k Q_k \\
&\leq \sum_{k=1}^{\infty} p_k Q_{k-1} \\
&= \sum_{k=1}^{\infty} Q_{k-1} - Q_k \\
&= Q_0 - \lim_{k \to \infty} Q_k \\
&\leq 1.
\end{aligned}
\tag{34}
$$

This ends the proof of Lemma B.10 and therefore proves Lemma B.5. $\qquad \square$

### B.4 PROOF OF LEMMA B.6

*Proof.* Consider the moment when a $x_t \in C_i^b$ arrives, and denote $s$ as the most recent moment ($s < t$) such that $x_s \in C_i^g$. According to the uniform permutation assumption from $Z$ to $\{x_t\}_{t=1}^T$, this $x_s$ can be any $z \in C_i^g$ with equal probability as $\Pr = \frac{1}{|C_i^g|} = \frac{2}{|C_i^*|}$. In the following, we analyze the expected loss $\mathbb{E}[l_t]$ by two cases:

(I) If $x_s \in C_i^g$ does exist before $x_t$ occurs. Denote $f_t^* := \arg\min_{f \in S_t} d(c_i^*, f)$ as the closest context to $c_i^*$ existed by the time $t$. Then we have:

$$
\begin{aligned}
\mathbb{E}[l_t | \{x_t\}_{t=1}^T] &= c \cdot \frac{\hat{d}_t(x_t, f_t)}{c} + d(x_t, f_t) \cdot (1 - \frac{\hat{d}_t(x_t, f_t)}{c}) \\
&\leq \hat{d}_t(x_t, f_t) + d(x_t, f_t) \\
&\leq \check{d}_t(x_t, f_t) + 2\Delta_t + \check{d}_t(x_t, f_t) + 2\Delta_t \\
&\leq 2\check{d}_t(x_t, f_t^*) + 4\Delta_t \\
&\leq 2d(x_t, f_t^*) + 4\Delta_t \\
&\leq 4d(x_t, c_i^*) + 4d(c_i^*, f_s^*) + 4\Delta_t.
\end{aligned}
\tag{35}
$$

Also, denote $\hat{f}_s := \arg\min_{f \in S_s} d(x_s, f)$ as the best existing context that can be matched to $x_s$ by the time $s$. Then we know that

$$
\begin{aligned}
\mathbb{E}[l_s | \{x_t\}_{t=1}^T] &\geq d(x_s, \hat{f}_s) \\
&\geq \frac{1}{2} d(c_i^*, \hat{f}_s) - d(x_s, c_i^*) \\
&\geq \frac{1}{2} d(c_i^*, f_s^*) - d(x_s, c_i^*).
\end{aligned}
\tag{36}
$$

Combining Equation (35) with Equation (36), we have

$$
\begin{aligned}
\mathbb{E}[l_t | \{x_t\}_{t=1}^T] &\leq 4d(x_t, c_i^*) + 4\Delta_t + 4 \cdot 2(\mathbb{E}[l_s | \{x_t\}_{t=1}^T] + d(x_s, c_i^*)) \\
&\leq 4d_t^* + 2\Delta_t + 8 \cdot \frac{2}{|C_i^*|} \cdot \Big( \sum_{s:x_s \in C_i^g} \mathbb{E}[l_s | \{x_t\}_{t=1}^T] + d_s^* \Big).
\end{aligned}
\tag{37}
$$

Again, the last line of Equation (37) comes from the i.i.d. assumption of $x_s$.

(II) If $x_s \in C_i^g$ does not exist before $x_t$ occurs, i.e. $x_r \in C_i^b, \forall r \leq t - 1$. According to the uniform permutation from $Z$ to $\{x_t\}_{t=1}^T$, this event happens with probability $\frac{2}{|C_i^*|}$. In this case, if $\hat{d}_t(x_t, f_t) \geq c$, then we suffer a cost $c$ at time $t$; otherwise $\hat{d}_t(x_t, f_t) < c$, and we either generate a new action (with cost $c$) or suffer an expected loss at $d(x_t, f_t) \leq \hat{d}_t(x_t, f_t) < c$. In a nutshell, the expected loss does not exceed $c$.

Combining with Case I and Case II above, we immediately get Equation (20). $\qquad\square$

## B.5   PROOF OF LEMMA B.8

*Proof.* Denote $\Delta_t := \Delta_t(x_t, f_t)$ for simplicity. In the following, we first reduce the summation of estimation error $\Delta_t$ to the regret of a $K(\leq T)$-arm linear bandit problem, up to constant factors. In fact, according to Lemma B.2, we know that $d(x_t, f) \leq \check{d}_t(x_t, f) \leq d(x_t, f)$. Since we select $f_t = \arg\min_{f \in S_t} \check{d}_t(x_t, f)$, we have

$$
\begin{aligned}
d(x_t, f_t^*) - 2\Delta_t &\leq d(x_t, f_t) - 2\Delta_t \\
&\leq \check{d}_t(x_t, f_t) \\
&\leq \check{d}_t(x_t, f_t^*) \\
&\leq d(x_t, f_t^*),
\end{aligned}
\tag{38}
$$

where $f_t^* := \arg\min_{f \in S_t} d(x_t, f)$ as the best existing context for $x$ in the current context library at time $t$. Therefore, the performance gap between $f_t$ and $f_t^*$ can be bounded as $d(x_t, f_t) - d(x_t, f_t^*) \leq 2\Delta_t$. On the other hand, since $d(x_t, f_t) = <w, \phi(x_t, f_t)>$ is a linear loss function, we consider $\phi(x_t, f)$ as the "context"[4] of each arm $f \in S_t$, and then we form a linear contextual bandit problem setting. Recall that $\Delta_t = \alpha \cdot \sqrt{\phi(x_t, f_t)^\top \Sigma_{t-1}^{-1} \phi(x_t, f_t)}$. According to Chu et al. (2011) Lemma 3 (which originates from Auer (2002) Lemma 13), we have

$$
\begin{aligned}
\sum_{t=1}^T \Delta_t = \sum_{t=1}^T \alpha \cdot \sqrt{\phi(x_t, f_t)^\top \Sigma_{t-1}^{-1} \phi(x_t, f_t)} \\
\leq \alpha \cdot 5\sqrt{(d^2)|\Psi_{T+1}| \log |\Psi_{T+1}|} \\
\leq 5\alpha\sqrt{d^2 T \log T}.
\end{aligned}
\tag{39}
$$

Here the second line is because the dimension of contexts are $d^2$ as $\phi(x_t, f) = Vec((x_t - f)(x_t - f)^\top) \in \mathbb{R}^{d^2}$, and the third line comes from the original definition of $\Psi_t$ as a subset of $[t-1]$. $\qquad\square$

## B.6   PROOF OF LOWER BOUND (THEOREM 5.5)

*Proof.* In order to prove the lower bound, we show the following facts

1. $OPT_o = \Omega(T^{\frac{d}{d+2}})$ according to the $K$-nearest-neighbors(K-NN) lower bound.

2. Any online facility location algorithm suffers at least $(2 - o(1))$-competitive-ratio, i.e. $ALG \geq (2 - o(1))OPT_h$.

In the following, we present two lemmas corresponding to the facts above.

**Lemma B.11** ($OPT_o$ lower bound). *We have $OPT_h \geq OPT_o \geq \Omega(T^{\frac{d}{d+2}})$.*

*Proof.* Denote $OPT_o(K) := \min_{S:|S|=K} c \cdot |S| + \sum_{t=1}^T \min_{f \in S} d(x_t, f) = K + T \cdot \min_{S:|S|=K} \frac{1}{T} \sum_{t=1}^T \min_{f \in S} d(x_t, f)$. This equals $T$ times $K$-nearest-neighbors (K-NN) loss plus $K$. According to Zador (1964) (i.e. Zador's Theorem in coding theory), the mean squared distance

---

[4]Here we denote this covariate as the *context* as it serves as an environmental description in the contextual bandits. We denote $f \in S_t$, which was denoted as a context in the library, an *arm* of this contextual bandits.

to the nearest codebook center in $\mathbb{R}^d$ space in $L_r$-norm is lower bounded by $\Omega(K^{-\frac{r}{d}})$. This is directly applicable to K-NN which effectively partitions points by their nearest neighbors. Hence, the quantization lower bound established by Zador's Theorem translates into a lower bound on K-NN's average squared loss. Therefore, we let $r = 2$ to fit in our setting, and then have

$$OPT_o = \min_{K \in [T]} OPT_o(K) = \Omega(c \cdot K + T \cdot K^{-\frac{2}{d}}) = \Omega(T^{\frac{d}{d+2}}), \tag{40}$$

where the last line is an application of Hölder's Inequality that $K + T \cdot K^{-\frac{2}{d}} \geq K^{\frac{\frac{2}{d}}{1+\frac{2}{d}}}(T \cdot K^{-\frac{2}{d}})^{\frac{1}{1+\frac{2}{d}}} = T^{\frac{d}{d+2}}$, and the equality holds at $K = T^{\frac{d}{d+2}}$. $\qquad \square$

**Lemma B.12** (Theorem 5.1 in Kaplan et al. (2023)). *Let $\mathcal{A}$ be an algorithm for online facility location in the i.i.d. model, then, the competitive ratio of $\mathcal{A}$ is at least $2 - o(1)$.*

Combining Lemma B.11 and Lemma B.12, we know that $REG = ALG - OPT_h \geq (2 - o(1) - 1)OPT_h \geq 0.5 OPT_o = \Omega(T^{\frac{d}{d+2}})$. This proves Theorem 5.5 $\qquad \square$

# C  THE CONFIDENCE BOUND OF NEURAL NETWORK FOR REGRESSION

In our real-world numerical experiments conducted in Section 6.2, we use a neural network to replace the linear regression estimator (i.e. the estimate of $W$ matrix) in the learning task on the Medical Q&A Dataset. In order to use the same framework as demonstrated in Algorithm 1, we wish to not only obtain point predictions but also the confidence bound of our predictions. In this section, we derive a general approach to achieve this quantification of NN output uncertainties.

Denote $\hat{y} = f_{\boldsymbol{\theta}}(\mathbf{x})$ as the neural network and the output prediction. One way to achieve this is by approximating the posterior distribution of the parameters $\boldsymbol{\theta}$ with a Gaussian distribution (a Laplace approximation), leveraging the fact that under a Gaussian prior and Gaussian likelihood, the resulting posterior can be treated via approximate conjugacy.

**Posterior approximation using Laplace:** Assume that we place an isotropic Gaussian prior on the parameters:

$$p(\boldsymbol{\theta}) = \mathcal{N}(\boldsymbol{\theta} \mid \mathbf{0}, s^2 \mathbf{I}), \tag{41}$$

and model the observations with Gaussian noise:

$$p(\mathbf{y} \mid \mathbf{x}, \boldsymbol{\theta}) = \mathcal{N}(\mathbf{y} \mid f_{\boldsymbol{\theta}}(\mathbf{x}), \sigma^2 \mathbf{I}). \tag{42}$$

By defining $\mathcal{L}(\boldsymbol{\theta}) = -\log p(\boldsymbol{\theta} \mid D)$, the Laplace approximation fits a Gaussian around the MAP solution $\hat{\boldsymbol{\theta}}_{\text{MAP}}$. Denoting the Gauss–Newton Hessian approximation by

$$\nabla_{\boldsymbol{\theta}}^2 \mathcal{L}(\hat{\boldsymbol{\theta}}_{\text{MAP}}) \approx \frac{1}{s^2}\mathbf{I} + \frac{1}{\sigma^2}\sum_{i=1}^n \nabla_{\boldsymbol{\theta}} f_{\hat{\boldsymbol{\theta}}_{\text{MAP}}}(\mathbf{x}_i)\,\nabla_{\boldsymbol{\theta}} f_{\hat{\boldsymbol{\theta}}_{\text{MAP}}}(\mathbf{x}_i)^{\top}, \tag{43}$$

we obtain the approximate posterior

$$\boldsymbol{\theta} \mid D \approx \mathcal{N}\!\left(\hat{\boldsymbol{\theta}}_{\text{MAP}}, \boldsymbol{\Sigma}\right) \quad \text{where} \quad \boldsymbol{\Sigma} = \left(\nabla_{\boldsymbol{\theta}}^2 \mathcal{L}(\hat{\boldsymbol{\theta}}_{\text{MAP}})\right)^{-1}. \tag{44}$$

**Confidence bound on $f_{\boldsymbol{\theta}}(\mathbf{x})$:** Since $\boldsymbol{\theta}$ is approximately Gaussian-distributed, a first-order (Delta-method) linearization gives

$$f_{\boldsymbol{\theta}}(\mathbf{x}) \approx f_{\hat{\boldsymbol{\theta}}_{\text{MAP}}}(\mathbf{x}) + \nabla_{\boldsymbol{\theta}} f_{\hat{\boldsymbol{\theta}}_{\text{MAP}}}(\mathbf{x})\,(\boldsymbol{\theta} - \hat{\boldsymbol{\theta}}_{\text{MAP}}). \tag{45}$$

Hence, the variance is:

$$\text{Var}[f_{\boldsymbol{\theta}}(\mathbf{x})] \approx \nabla_{\boldsymbol{\theta}} f_{\hat{\boldsymbol{\theta}}_{\text{MAP}}}(\mathbf{x})\, \boldsymbol{\Sigma} \left[\nabla_{\boldsymbol{\theta}} f_{\hat{\boldsymbol{\theta}}_{\text{MAP}}}(\mathbf{x})\right]^{\top}. \tag{46}$$

**Online update via Sherman–Morrison:** When data arrives sequentially, we can maintain and update the Hessian approximation (or its inverse) via a rank-1 update procedure:

$$A_t \;=\; \frac{1}{s^2}\mathbf{I} \;+\; \frac{1}{\sigma^2} \sum_{i=1}^{t} \nabla_{\boldsymbol{\theta}} f_{\hat{\boldsymbol{\theta}}}(\mathbf{x}_i)\,\nabla_{\boldsymbol{\theta}} f_{\hat{\boldsymbol{\theta}}}(\mathbf{x}_i)^{\top}, \tag{47}$$

and

$$A_{t+1} \;=\; A_t \;+\; \frac{1}{\sigma^2} \left[\nabla_{\boldsymbol{\theta}} f_{\hat{\boldsymbol{\theta}}}(\mathbf{x}_{t+1})\right] \left[\nabla_{\boldsymbol{\theta}} f_{\hat{\boldsymbol{\theta}}}(\mathbf{x}_{t+1})\right]^{\top}. \tag{48}$$

Using the Sherman–Morrison formula, the inverse $A_{t+1}^{-1}$ can be efficiently computed in $O(d^2)$:

$$A_{t+1}^{-1} = A_t^{-1} \;-\; \frac{\frac{1}{\sigma^2}\,A_t^{-1}\,\mathbf{g}_{t+1}\,\mathbf{g}_{t+1}^{\top}\,A_t^{-1}}{1 + \frac{1}{\sigma^2}\,\mathbf{g}_{t+1}^{\top} A_t^{-1}\,\mathbf{g}_{t+1}} \quad \text{where} \quad \mathbf{g}_{t+1} := \nabla_{\boldsymbol{\theta}} f_{\hat{\boldsymbol{\theta}}}(\mathbf{x}_{t+1}). \tag{49}$$

Thus, at any point in time, we can maintain an approximate posterior inverse Hessian and readily compute a confidence bound on $f_{\boldsymbol{\theta}}(\mathbf{x})$ by plugging $A_t^{-1}$ into the variance formula above.

