# OpenReview forum: "Online Decision Making with Generative Action Sets"
_ICLR.cc/2026/Conference — ICLR 2026 Poster_

### Official Review · Reviewer_GG79 · 2025-10-30

**Soundness:** 3
**Presentation:** 3
**Contribution:** 2
**Rating:** 6
**Confidence:** 4

**Summary:**

This paper studies an online learning problem with an expanding action space, motivated by generative AI systems that can create new actions on demand at a cost. The authors introduce the create-to-reuse framework, in which an agent decides at each time step whether to select an existing action or pay a one-time cost to generate a new, reusable action.
To solve this problem, the authors propose a doubly-optimistic algorithm that employs Lower Confidence Bounds (LCB) for action selection among existing options, and Upper Confidence Bounds (UCB) for deciding whether to create new actions.
This approach is theoretically shown to achieve a sublinear regret bound of O\left(T^{\frac{d}{d+2}} d^{\frac{d}{d+2}} + d \sqrt{T \log T}\right),
which matches an  Ω(T^{d/(d+2)}) lower bound—making it optimal with respect to the time horizon T.
Empirical validation includes both synthetic data (to confirm regret rates) and real healthcare question-answering datasets (to demonstrate tradeoffs between creation cost and response mismatch). The proposed algorithm consistently outperforms fixed-probability baselines in terms of total loss and generation-quality efficiency.

**Strengths:**

1.  Introduces an important and timely extension of online decision-making- expanding the action space through generative capabilities.
2.  The double-optimism principle (LCB for reuse, UCB for creation) is conceptually appealing and well-motivated by the underlying tradeoffs.
3.  Provides nontrivial analysis, including tight regret bounds and a matching lower bound.
4. The healthcare Q&A examples effectively illustrate the value of balancing creation cost with reuse benefits.
5. Real-data results show meaningful gains and smooth generation-quality tradeoffs.

**Weaknesses:**

1. The algorithm’s  worst-case complexity O(d^4 T^2) is prohibitive for high-dimensional embeddings (e.g., large language model contexts).
   The paper proposes approximate implementations (linear or neural variants) but does not provide theoretical guarantees for them.

2.  The  quadratic loss model d(x,f) = (x-f)^\top W(x-f) may not generalize to real semantic spaces.

3.  Evaluation focuses mainly on healthcare QA; additional domains (e.g., recommendation systems or dynamic pricing) would strengthen claims of generality.

4.  The probabilistic decision rule for creation (based on ( UCB/c )) could be analyzed more thoroughly-what happens if the stochastic component is removed or tuned differently?

5.  Links to active learning, bandits with knapsacks, and growing action spaces (Farquhar et al., 2020) are discussed in the appendix, but not deeply integrated into the main narrative.

**Questions:**

1. How critical is the quadratic assumption on mismatch loss? Would kernelized or neural distance functions maintain similar regret behavior?

2. Can the proposed framework handle context-dependent creation costs c(x_t)? How would this affect the theoretical analysis?

3. Could approximate nearest neighbor search or library pruning reduce complexity while preserving regret guarantees?

4. The create-to-reuse tradeoff resembles selective labeling. Could active learning techniques (e.g., expected information gain) enhance the creation policy?

5. How would the method extend to stochastic or partially observable generation outcomes (as in LLM-based action creation)?

---

> ### Author Response · Authors · 2025-11-18
>
> Thanks for your supportive and insightful feedback! We address your concerns and questions as follows:
>
> **1, Computational Complexity and Theoretical Guarantee:**
>
> We acknowledge the computational challenge and theory-practice gap. The full quadratic model incurs $O(d^4)$ per-step cost (computing confidence bounds for all contexts in $S_t$), which is indeed prohibitive for high-dimensional embeddings.
>
> For the rank-1 (squared linear) model, we do have theoretical guarantees: the model $W = \theta\theta^\top$ reduces per-step complexity to $O(d)$ while remaining a special case of our quadratic framework. The $O(T^{d/(d+2)})$ creation-rate bound applies directly, as this rate arises from covering geometry rather than parametric form. The regret bound $O(T^{d/(d+2)} d^{d/(d+2)} + d\sqrt{T log T})$ holds, with potential improvement to the second term under known rank-1 structure.
>
> For the neural network model, we agree there is no end-to-end theoretical guarantee. However, we maintain the algorithmic principle (LCB-reuse, UCB-creation) and rigorously derive confidence bounds $\Delta_t(x,f)$ in Appendix C via Laplace approximation of the Gaussian posterior. This ensures the doubly-optimistic mechanism is properly implemented, though analyzing the full regret guarantee with neural approximators remains important future work.
>
> **2, Quadratic Loss Assumption and Generalization:**
>
> The quadratic model is a preliminary choice, as our primary contribution is the problem framework and doubly-optimistic methodology rather than a definitive distance model. That said, the quadratic form is reasonable for semantic embeddings: modern embeddings (e.g., from transformers) are specifically trained to make Euclidean distance meaningful, and W allows reweighting dimensions by task relevance.
>
> **Regarding regret behavior**: our bound decomposes into $O(T^{d/(d+2)} d^{d/(d+2)})$ (create-vs-reuse geometry) + $O(d\sqrt{T \log T})$ (distance estimation error). The quadratic assumption primarily affects the second term—the first arises from covering/packing arguments that depend on metric properties (smoothness, dimension), not parametric form. Therefore, we expect similar $O(T^{d/(d+2)})$ geometric rates for broader function classes (Hölder-smooth, kernel-based metrics), though the estimation term would depend on the function class complexity (e.g., RKHS norms, eluder dimension). Some possible generalization approaches:
>
> (1) Hölder-smooth non-parametric losses: For loss functions $d(x,f)$ with Hölder smoothness, the algorithm can be extended via discretization of the parameter space. By maintaining confidence sets over discretized hypotheses and refining the partition over time, the doubly-optimistic principle carries through, though analyzing the regret-complexity tradeoff (error from discretization vs. variance from learning) is technically involved. We conjecture that the regret would become $O(T^{\frac{d}{d+2}}d^{\frac{d}{d+2}} + T^{\frac{\alpha + d}{2\alpha + d}})$ where the first term is the same and the second term comes from the regret rate of continuum-armed bandits with $\alpha^{th}$-order smoothness in $d$-dimension space.
>
> (2) Kernel-based metrics (e.g., Gaussian kernel): Similar to Bayesian Optimization, we can model $d(x,f)$ as a Gaussian Process and derive uncertainty quantification from the posterior variance. The LCB/UCB construction would use GP confidence bounds instead of parametric ones. Cosine similarity could be handled similarly by embedding into a reproducing kernel Hilbert space (RKHS).
>
> **3, Evaluation on other domains:**
>
> Our empirical evaluation focuses on two healthcare Q&A datasets, which reflects a core motivation for this work—intelligent FAQ reuse is a practically important problem in healthcare information systems where action creation (with domain expert review) is genuinely costly. While other applications exist (e.g., personalized advertisement, content recommendation, as discussed in the Introduction), our primary contribution is establishing the novel problem formulation, theoretical framework with provable guarantees, and demonstrating feasibility on real-world data. Broader empirical validation across diverse domains is valuable future work building on this foundation, including (1) Personalized content recommendation (as we mentioned in the Introduction, creating new articles/videos versus reusing existing content with some user-preference mismatch), (2) Drug discovery (reuse a known scaffold/compound vs create and synthesize a new candidate), and (3) Educational content delivery (generating tailored lessons vs. adapting existing materials).
>
> (to be continued in the next **Official Comment** block)

---

> ### Author Response · Authors · 2025-11-18
>
> (continuing from previous block)
>
> **4, Methods (like ANN search or library pruning) to reduce complexity:**
>
> We add a paragraph in Appendix A.2 where we discuss mitigation strategies including approximate nearest-neighbor search for action selection and de-activation methods that maintain a frequently-used action subset, both of which can reduce per-step complexity substantially in practice. With a careful high-probability bound being imposed, we conjecture that this will not affect the dominant term of our regret bound, which would be an interesting future work to explore.
>
> **5, Probabilistic Creation Rule:**
>
> The Bernoulli randomization (creating with probability $UCB_t(x_t, f)/c$) serves a precise purpose: it makes the expected creation cost and expected mismatch loss directly comparable when deciding whether to create or reuse, enabling principled tradeoff balancing in expectation.
>
> The UCB (upper confidence bound) strategy embodies optimism about newly created actions—we overestimate their future reuse value to encourage exploration of the action space, balancing the exploration-exploitation tradeoff inherent in building a reusable library. In contrast, using an LCB (lower confidence bound) would be overly pessimistic and perform even worse than the Fixed-P baseline frequently.
>
> Regarding deterministic rules: replacing Bernoulli sampling with a fixed threshold $c'$ is theoretically possible under i.i.d. context arrivals, but would require more sophisticated analysis (handling worst-case orderings, ensuring sufficient library diversity) without meaningful practical or theoretical benefit. The randomized rule simplifies both implementation and analysis while achieving near-optimal regret.
>
> **6, Connection to selective labeling and other active learning methods:**
>
> Thank you for pointing out this connection. While there is surface similarity to selective labeling, the core objective differs fundamentally: active learning optimizes information gain to improve model accuracy with minimal labeling cost, whereas our setting involves an immediate economic tradeoff between creation cost $c$ and expected mismatch loss when reusing.
>
> Our creation decision is driven by this cost-reward balance (via the UCB/c rule) rather than epistemic value. That said, incorporating expected information gain as an auxiliary signal is an interesting direction: prioritizing creation when it would most improve distance estimates could accelerate learning, though balancing this with immediate cost-effectiveness presents nontrivial challenges.
>
> Regarding related work placement: we relegated active learning, bandits-with-knapsacks, and growing action spaces to the appendix primarily due to page limits during submission. Upon acceptance, we will integrate the most relevant connections into the main narrative.
>
> **7, Stochastic or partially observable generation:**
>
> This is an important question, especially for LLM-based applications where generation is inherently stochastic (e.g., sampling with temperature). In such settings, calling $\mathcal{A}(x_t)$ multiple times yields different outputs, and we may not fully observe the true quality or representation of the generated action until it's deployed. Just as with a simple imperfect oracle, our algorithm, which operates in the representation space, would perceive this as noise or bias in the observed representation. When we prompt an LLM to create an action for a state $x_t$, the stochastic nature of the generation means the resulting action might correspond to an effective state $x'$, and we would observe $x_t$ as a noisy proxy.
>
> The fundamental challenge here is that the mismatch loss $l_t(\text{new})$ is no longer a fixed unknown value but a random variable, drawn from a distribution conditioned on the prompt. This transforms the 'create' decision into a risk-management problem: the agent must weigh the fixed cost $c$ against the expected mismatch loss from the new action, rather than a deterministic one. From a theoretical standpoint, the regret analysis would now depend on the properties of this generation distribution. If the variance of the mismatch loss is bounded, we might be able to adapt our analysis by incorporating this expected error. However, if the distribution is heavy-tailed or adversarial (for instance, if the LLM can 'hallucinate' and produce a completely irrelevant action), the potential for large, unbounded single-step losses could invalidate our current regret bounds. Modeling and adapting to these complex, generative oracles is a critical and exciting direction for future research.
>
>
> ---------
> We hope these have addressed your concerns. Please let us know if there are any other questions. Thanks!

---

### Official Review · Reviewer_pGC5 · 2025-10-30

**Soundness:** 3
**Presentation:** 4
**Contribution:** 3
**Rating:** 6
**Confidence:** 3

**Summary:**

This paper studies a new bandit framework where a decision maker receives at each round a context $x_t$, and decides whether to (i) pick an optimal action $\mathcal{A}(f)$ corresponding to an already-seen context $f\in S_t$, or (ii) incur a fixed cost $c>0$ and generate a new action $\mathcal{A}(x_t)$ specifically tailored to $x_t$, which becomes available for future rounds. This problem is mainy motivated  by healthcare QA systems. The authors propose a doubly-optimist algorithm to  solve this problem, and derive an optimal-in-T upper bound on the regret incurred by their algorithm. They then run two sets of experiements: one exactly implements the algorithm in a small dimension synthetic data setup; the other implements an approximate version of the algorithm (where the distance is predicted linearly) one actual QA data.

**Strengths:**

1) The paper is very-well written and pedagogical. The authors did a great job presenting their setting and algorithm in a simple and intuitive way, without sacrificing mathematical soundness.
2) The setting is interesting and is well motivated from a practical point of view (healthcare QA)
3) The theoretical analysis is complete, thanks to a lower-bound complements the upper-bound on the regret incurred by the introduced algorithm. The convergence rate is very slow (exponential dependence on the dimension, almost linear dependence on T for large d), however this has to do with the inherent difficulty of the problem (as suggested by the lower bound).

**Weaknesses:**

1) I am not entirely convinced by the way the loss incurred by the algorithm is modeled. Indeed, the paper assumes that the loss incurred at each round is given by a (noisy) distance generated by $W$ between the actual context $x_t$ and the context chosen to generate the action. This is a quite restrictive modelling. In particular I would expect W to depend on $x_t$, so the loss would be $(x_t - f)^\top W(x_t) (x_t -f)$. Indeed, answering to a question $f$ instead of the actual $x_t$ is likely to be more harmful when the user asks about cancer than about superficial wound for instance. By the way, this formulation would naturally arise by considering the second order Taylor expansion of the excess risk $h(x_t, \mathcal{A}(f)) - h(x_t, \mathcal{A}(x_t))$ for some decision loss $h$.
2) The authors implement the algorithm for which they derive guarantees only on a very toyish example, and use a simplified version on the actiual QA dataset. Altough I understand that the dependence on $d$ makes the cost of an actual implementation prohibitive, it is quite unfortunate not to have a demonstration of the effectiveness of the algorithm on actual data, and not to have a theoretical guarantee of the simplified algorithm used in practise.

**Questions:**

1) Is it possible to extend the analysis to the case where $W$ depends on the context $x_t$ ?
2) Are there other motivations for the problem studied in this paper?

---

> ### Author Response · Authors · 2025-11-18
>
> Thanks for your supportive and insightful feedback! We address your concerns and questions as follows:
>
> **1, Context-Dependent Loss Matrix $W(x_t)$:**
>
> This is very thoughtful. Since this work mainly aims to introduce the create-to-reuse problem framework and our doubly-optimistic methodology, the fixed-$W$ model assumed here is a preliminary approach for a rigorous derivation of the end-to-end theoretical guarantee. While the current model is able to measure the difference in reality (as the key information is already captured in the embedding vector $x_t$), allowing $W$ to depend on $x_t$ is indeed an interesting and valuable extension that would capture asymmetric mismatch costs. As shown in your excellent example, answering a cancer question with a wound-treatment response may be more harmful than the reverse, which a context-dependent $W(x_t)$ could model naturally.
>
> However, without additional structure, learning $W(x_t)$ becomes fundamentally challenging: for each newly observed context $x_t$, we would have no prior data about $W(x_t)$ specifically, making prediction infeasible. To make this extension learnable, we would need to parametrize $W(x_t)$ with shared structure across contexts. Possible approaches include:
>
> (a) Factorized form: $W(x_t) = W_0 + \sum_i x_t[i]·W_i$, where the coefficient $\{W_i\}$’s are learnable.
>
> (b) Low-rank bilinear model: $W(x_t) = \sum_k (a_k^T x_t)\cdot B_k$.
>
> (c) Smoothness or kernel-based method.
>
> All these three models allow a finite covering of the hypothesis space, but they still present distinct theoretical and computational challenges (e.g., identifiability, sample complexity scaling with context dimension). This is an exciting direction for future work that would significantly broaden the modeling flexibility and better capture real-world asymmetries as you have highlighted.
>
> **2, The theory-implementation gap:**
>
> Thank you for this important observation. We acknowledge the disconnect between our theoretical analysis (quadratic loss) and high-dimensional implementation (rank-1/neural network models) and will make this limitation more explicit.
>
> **(1) On the quadratic model**: While computationally expensive in high dimensions, it is not merely "toyish" as it arises naturally in two important settings: (i) metric learning on semantic embeddings, where learned $W$ reweights dimensions by task-relevance, and (ii) local approximations of excess risk $h(x, A(f)) - h(x, A(x))$, whose second-order Taylor expansion yields a quadratic form with Hessian $W$.
>
> **(2) For the rank-1 (squared linear) model** used in experiments, theoretical guarantees do extend: this is a computationally efficient special case where $W = \theta\theta^\top$. The $O(T^{d/(d+2)}d^{d/(d+2)})$ regret bound still holds, as this rate arises from create-vs-reuse geometry (a covering/facility-location effect), not parametric form. The estimation error term may even improve from $O(d\sqrt{T \log T})$ to $O(d \log T)$ if we have the pre-knowledge of this rank-1 structure ahead of time.
>
> **(3) For the neural network model**, we maintain the algorithmic principle (LCB-reuse, UCB-creation) while using NNs for expressiveness. We do not have end-to-end regret guarantees here, but we added a Appendix C that rigorously derives confidence bounds $\Delta_t(x,f)$ via Gaussian posterior approximation, ensuring the doubly-optimistic principle is properly implemented. We agree this is a limitation and view extending theoretical analysis to broader function classes (Hölder-smooth, kernel-based) as important future work.
>
> **3, Other Motivations of this study:**
>
> Yes, several domains exhibit this create-to-reuse structure. (1) As we mentioned in the Introduction, Personalized content recommendation (creating new articles/videos versus reusing existing content with some user-preference mismatch). (2) Drug discovery (reuse a known scaffold/compound vs create and synthesize a new candidate). (3) Educational content delivery (generating tailored lessons vs. adapting existing materials).  All of them involve costly creation decisions balanced against imperfect reuse. The framework naturally applies wherever building a reusable library incrementally is more efficient than creating from scratch each time.
>
> ---
> We hope these have addressed your concerns. Please let us know if there are any other questions. Thanks!

---

> > ### Comment · Reviewer_pGC5 · 2025-11-26
> >
> > I thank the authors for their very thorough answer.
> >
> > Regarding the possibility of making $W$ depend on $x_t$, I think a low-rank approximation is indeed an interesting direction. It would also be valuable to look at quasi-Newton methods, which are specifically tailored to compute low-rank approximations of the Hessian on-the-fly (see for instance the BFGS approximate). In any way, I think that $W$ being fixed (which corresponds to the cost function $h$ being an exact quadratic form) should be discussed in the main text.
> >
> > Regarding the theory/experiment gap, I understand that the computational obstacle is hard to overcome. This is an intrisiquic weakness of the method, which is nonetheless clearly and honestly discussed by the authors in the main text.
> >
> > Regarding other motivations for the method, I thank the authors for reminding the examples they already put in introduction.
> >
> > All in all, I am willing to keep my score unchanged: in spite of the cost modelling being a bit simplistic and the experiments not being perfectly adapted to the theory, the paper is innovative and well motivated. I therefore think it deserves to be accepted.

---

### Official Review · Reviewer_yeWb · 2025-11-01

**Soundness:** 3
**Presentation:** 3
**Contribution:** 3
**Rating:** 6
**Confidence:** 2

**Summary:**

The authors of the paper propose a method of reusing annotated actions in an online decision making framework. Specifically, the authors propose an algorithm that takes into account the cost of adding an action with cost $c$ (annotated answer in an FAQ setting) into a context library $S_t$ over reusing an already existing action which, might lead a loss due to the sub-optimal nature of the already existing answer. To reduce the overall computational complexity, the For theoretical purposes to calculate the regret bounds , the loss is approximated as a weighted squared distance, though in the empirical results, they use a squared linear model or a neural network (NN).

Notation:
- Input context variable $x_t \in \mathbb{R}^d$ i.e. vector embedding for questions in FAQ setting
- $S_t:= \{x_t, a_t\}$ is a context library of vetted FAQs that is stored as a hash map.
- $d(x,f)$ is the distance function which models the loss.

**Strengths:**

Strengths:
- The paper address an important aspect of a real world problem i.e. when does one need to annotate new data vs. use existing data. Given that data annotation is a fairly laborious and expensive process, this would be a prudent approach proposed by the authors.
- The approach is fairly simple and approachable. The algorithm in Algorithm 1 was easy to understand along with the reasoning about it's time complexity (though I'm not too convinced about using NNs to help with that). The algorithm also makes sense in having a LCB and UCB defined for the overall loss from a theoretical sense, but I might see issues here of it being practical.

**Weaknesses:**

Weakness:
- The use of a NN to approximate the distance function to reduce computational complexity seems a little counter intuitive. Typical NNs would be sufficiently large that it might render this moot.
- There is no study on how the size of the context library might impact the algorithm (like a cold start problem). It would also be good to understand if this framework leads to long term stability of needing an oracle.
- I did not fully evaluate the authors theoretical claims, but I think some of their empirical results are not very convincing. Its hard to gauge the performance of the algorithm from 2 real world datasets.
- I also think that from a practical standpoint, it would be hard for practitioners to tie in the relation between the distance metric and cost (the notion of a *total loss*)

**Questions:**

Questions:
- Section 6.2 line 432: "cost parameter $c$ from 0 to 100 for our algorithm ..." : Can you explain why you do this sweep across? Also, what does this cost mean from an intuitive sense? How does this tie back into the the *total loss*
- I might have also missed this, but could you explain how you estimate/train the distance function for both the datasets in section 6 i.e. the squared liner model and NN mentioned in section 4.

---

> ### Author Response · Authors · 2025-11-18
>
> Thanks for your supportive and insightful feedback! We address your concerns and questions as follows:
>
> **1, The Simplicity of Neural Network Computation:**
>
> Thank you for this clarification question. The neural network extension is for *increased expressiveness* (handling complex, non-quadratic mismatch functions), not computational efficiency compared to the linear model. Indeed, NNs are more computationally expensive than quadratic models. The "simplicity" we refer to is in the *uncertainty quantification method*: rather than performing rigorous Bayesian inference over NN parameters (which would be computationally intractable), we use Laplace approximation to estimate a Gaussian posterior combined with Gauss-Newton approximation of the Hessian to construct confidence bounds. Details are provided in Appendix C.
>
> We acknowledge this is a practical tradeoff: We sacrifice end-to-end theoretical guarantees for the ability to handle more realistic loss functions. The approximations make NN-based uncertainty quantification feasible, and it is even more efficient than the original quadratic-loss form if $D\leq d^2$ where $D$ denotes the total number of parameters of NN. For applications requiring non-quadratic metrics (e.g., cosine similarity in embedding spaces), this added cost is justified by improved model fidelity.
>
> **2, Library Size and Cold-Start:**
>
> **(1)** On library size scalability: Computing LCB for every $f ∈ S_t$ implies per-step cost $O(|S_t|·d²)$. While our theory shows expected library growth scales as $O(T^{d/(d+2)})$, in high-dimensional embeddings this exponent approaches $1$, so near-linear growth can occur for small creation costs $c$. We have added empirical analysis to Appendix A.2 showing $|S_t|$ w.r.t. $t$ for various $c$ values, and showed that the growth rate is strongly governed by $c$, with larger costs yielding much slower growth. We also discuss mitigation strategies including approximate nearest-neighbor search for action selection and de-activation methods that maintain a frequently-used action subset, both of which can reduce per-step complexity substantially in practice.
>
> **(2)** On long-term oracle dependency: The sublinear $O(T^{d/(d+2)})$ creation rate implies oracle calls become increasingly rare over time—the algorithm asymptotically approaches self-sufficiency in stationary environments as the library achieves adequate coverage. However, in non-stationary settings with concept drift or evolving context distributions, continuous (though infrequent) oracle access may remain necessary to adapt to new regions of the context space.
>
> **3, Empirical Evaluation Scope:**
>
> We acknowledge our empirical evaluation focuses on two healthcare Q&A datasets. This reflects a core motivation for this work—intelligent FAQ reuse is a practically important problem in healthcare information systems where action creation (expert review) is genuinely costly. While other applications exist (e.g., personalized advertisement, content recommendation, as discussed in the Introduction), our primary contribution is establishing the novel problem formulation, theoretical framework with provable guarantees, and demonstrating feasibility on real-world data. Broader empirical validation across diverse domains is valuable future work building on this foundation.
>
> **4,  Practical Calibration and Cost Sweep (addressing both questions):**
>
> Intuitively, $c$ represents the cost of generating a new action and directly appears in the total loss as $Total\ loss = \sum_{t}d(x_t, f_t) + c \cdot |S_t|$. Therefore, $c$ controls the tradeoff between mismatch loss (reusing suboptimal actions) and creation cost (generating new actions). In practice, $c$ can be:
>
> (1) Domain-informed: Concrete costs like API cost, expert review time, synthesis expenses, etc.
>
> (2) Tunable hyper-parameter: Adjusted to achieve desired generation-reuse balance (like regularization in ML).
>
> **Why sweep c from 0 to 100:** Our goal is to compare *methods*, not to optimize $c$. Different $c$ values represent different tradeoff preferences under the same method. By sweeping $c$ (and similarly sweeping the probability $P$ for the Fixed-P baseline), we demonstrate *Pareto* *optimality* as our algorithm achieves superior generation-mismatch tradeoffs across the entire spectrum, dominating baselines regardless of the practitioner's cost preference. This provides comprehensive evidence of algorithmic superiority beyond any single operating point.
>
> (to be continued in the next **Official Comment** block)

---

> ### Author Response · Authors · 2025-11-18
>
> (continuing from previous block)
>
> **5, Distance Function Training:**
>
> Both models are trained online using observed feedback triples $(x_t, f_t, l_t)$ accumulated over time, where $l_t = 0$ if $f_t = x_t$ or $l_t=d(x_t, f_t) + N_t$ otherwise. For the squared linear model where $W=\theta\theta^\top$, we use linear regression to estimate the parameter $\theta$, with analytical formulas providing both the predictor and standard deviation for confidence bound construction. For the neural network, we use stochastic gradient descent (SGD) on the accumulated feedback to train the network, with confidence bounds derived from an approximation of the Gaussian posterior variance (please kindly find more details in Appendix C).
>
> -----------------------------
> We hope these have addressed your concerns. Please let us know if there are any other questions. Thanks!

---

### Official Review · Reviewer_WJ1K · 2025-11-04

**Soundness:** 3
**Presentation:** 3
**Contribution:** 3
**Rating:** 8
**Confidence:** 2

**Summary:**

The paper presents a novel sequential decision-making problem, where a decision maker has to learn to either pick a previously known action (exploitation/exploration) or generate a new one by paying a fixed cost (creation). The authors provide the first (doubly-optimistic) algorithm to solve this problem that is shown to achieve sub-linear regret, under some assumptions on the loss incurred. Further, the authors evaluate the performance of their approach on both synthetic and realistic datasets.

**Strengths:**

_(Disclaimer: this is not exactly my area of expertise. I skimmed through the appendices and proofs, but I did not check them too deeply.)_

The paper tackles an interesting and timely problem, which I believe has several interesting ramifications and implications for several human-AI decision-making scenarios. The paper is clear in its exposition, and all the relevant assumptions are duly explained and motivated. Furthermore, the related work is described comprehensively (both in the main text and appendix), enabling a clear contextualization of the paper's proposed method. Overall, I enjoyed reading it.

**Weaknesses:**

I do not see any fundamental weakness in this work, based on my limited expertise with multi-armed bandit settings. The paper is clear in explaining the weaknesses (e.g., assumptions) of the proposed approach.

I can only comment that some assumption feels very synthetic. For example, the quadratic loss function $d(x_t, f_t)$, while justified for tractability, limits generality and may underrepresent practical similarity measures (e.g., cosine or kernel-based metrics). Further, context-dependent or stochastic creation costs would also be interesting. Looking at practical applications, I would expect to measure the efficacy (regret) of Algorithm 1 in solving a certain decision-making task (e.g., curing a patient). Your algorithm would be oblivious to the fact that a pre-registered question is wrong, or it does not work for a given context (the same reasoning applies to generated actions).

**Questions:**

- How would you extend your algorithm to consider more complex notions of loss? For example, a probabilistic loss dependent on the action/context pair (e.g., efficacy of a treatment for a patient $x_t$).
- Would the regret-proof strategy still work under time- or context-dependent creation costs $c$? How do you envision your strategy to change?

---

> ### Author Response · Authors · 2025-11-18
>
> Thanks for your strong support and insightful feedback! We address your concerns and questions as follows:
>
> **1, Extension to more complex notions of loss:**
>
> Thank you for this interesting question. We have discussed extensions beyond quadratic loss in Appendix A.2 (see the paragraph of “Non-Parametric and Neural Function Approximation.”), and we're glad to elaborate further here.
>
> (1) Hölder-smooth non-parametric losses: For loss functions $d(x,f)$ with Hölder smoothness, the algorithm can be extended via discretization of the parameter space. By maintaining confidence sets over discretized hypotheses and refining the partition over time, the doubly-optimistic principle carries through, though analyzing the regret-complexity tradeoff (error from discretization vs. variance from learning) is technically involved. We conjecture that the regret would become $O(T^{\frac{d}{d+2}}d^{\frac{d}{d+2}} + T^{\frac{\alpha + d}{2\alpha + d}})$ where the first term is the same and the second term comes from the regret rate of continuum-armed bandits with $\alpha^{th}$-order smoothness in $d$-dimension space.
>
> (2) Kernel-based metrics (e.g., Gaussian kernel): Similar to Bayesian Optimization, we can model $d(x,f)$ as a Gaussian Process and derive uncertainty quantification from the posterior variance. The LCB/UCB construction would use GP confidence bounds instead of parametric ones. Cosine similarity could be handled similarly by embedding into a reproducing kernel Hilbert space (RKHS).
>
> (3) On probabilistic/stochastic losses (your treatment efficacy example): In our current setting, mismatch loss is a combination of deterministic function $d(x,f)$ with an observational iid subGaussian noise. This could partially capture the stochastic outcomes of treatment efficacy (the same (patient, treatment) pair yields different results due to inherent randomness), but it is not proper to make iid and subGaussian assumptions (since the probabilistic model is different from an add-on noise). We expect the algorithm framework still works, but since our regret analysis requires concentration of those subGaussian martingales, this change might affect the regret rate.
>
> **2, Dynamic/Context-Dependent Creation Costs:**
>
> Dynamic creation costs $c_t$ or context-dependent costs $c(x_t)$ would indeed better model realistic scenarios (e.g., drug synthesis costs varying with molecular complexity, content moderation costs varying with topic sensitivity). However, this extension introduces substantial algorithmic challenges—as evidenced by the variant-cost online facility location literature, where even achieving constant competitive ratios becomes impossible under adversarial cost sequences. We provide a detailed discussion of this extension and its technical challenges in Appendix A.2 (see the "Dynamic and Context-Dependent Creation Cost" paragraph).
>
> **3, Action Validity Concern:**
>
> That is a very insightful point. Our framework does operate under the assumption that all functions, whether pre-registered or newly generated, are valid and well-formed. The algorithm is designed to learn which function is most effective based on performance (i.e., minimizing loss), but it does not have a built-in mechanism to semantically evaluate a function's logic or flag it as inherently "wrong" for a given context before it is applied. Incorporating such a validation or safety layer, which could potentially prune or discard functions that are nonsensical or harmful, would be a critical extension for robust real-world deployment.
>
> --------------------
> We hope these have addressed your concerns. Please let us know if there are any other questions. Thanks!

---

> > ### Comment · Reviewer_WJ1K · 2025-11-27
> >
> > I would like to thank the authors for the comments and clarifications. I have no further questions to ask. After reading the other reviews, I am inclined to maintain my score, since I believe it is a solid contribution (given my limited confidence on the topic).

---

### Official Review · Reviewer_uKET · 2025-11-06

**Soundness:** 4
**Presentation:** 4
**Contribution:** 4
**Rating:** 8
**Confidence:** 4

**Summary:**

This paper introduces an online decision-making problem relevant to modern generative AI systems, where an agent can dynamically expand its action set. At each time step, the agent observes a context $x_t$ and faces a choice: (1) select an existing action $f$ from its library $S_t$ and incur a mismatch loss $d(x_t, f)$, or (2) pay a fixed cost $c$ to generate a new, context-specific action $\mathcal{A}(x_t)$ which provides zero mismatch loss and is then permanently added to the library $S_{t+1}$. This frames a novel "create-to-reuse" problem, which the authors identify as a triangular trade-off among exploitation, exploration, and creation.

To solve this, the authors propose a **Doubly-Optimistic Algorithm**. This algorithm first identifies the best *existing* action $f_t$ by using a Lower Confidence Bound (LCB) on the mismatch loss (balancing exploitation and exploration). It then makes a stochastic decision on whether to *create* a new action based on the Upper Confidence Bound (UCB) of $f_t$'s loss, $\hat{d}_t(x_t, f_t)$, opting to create with probability $\min(1, \hat{d}_t(x_t, f_t)/c)$.

The paper's main contributions are:
1.  The "create-to-reuse" problem formulation.
2.  The novel Doubly-Optimistic Algorithm.
3.  A rigorous theoretical regret analysis, which assumes the loss $d(x,f)$ is an unknown quadratic function $d(x,f) = (x-f)^\top W (x-f)$. The algorithm achieves an expected regret of $O(T^{\frac{d}{d+2}}d^{\frac{d}{d+2}} + d\sqrt{T \log T})$.
4.  A matching information-theoretic lower bound of $\Omega(T^{\frac{d}{d+2}})$ (with respect to $T$), proving the algorithm's optimality.
5.  Empirical validation on synthetic data (which verifies the theoretical regret rate) and on two real-world healthcare Q&A datasets (which demonstrates superior cost-quality trade-offs compared to a fixed-probability baseline).

**Strengths:**

* **Novel and Timely Problem:** The "create-to-reuse" formulation is a significant and timely contribution. It accurately models a new challenge posed by generative AI, where high-quality, custom actions (like detailed Q&A responses) are now possible to create on-the-fly, but at a non-trivial cost. This is a valuable and natural extension of the classic contextual bandit framework.
* **Elegant Algorithm Design:** The "doubly-optimistic" principle is an intuitive and elegant solution to the new exploration-exploitation-creation trade-off. Using an LCB for *action selection* (to manage uncertainty among *known* actions) while using a UCB-based probability for *action creation* (to optimistically value the benefit of creating a new action) is a clever method for decomposing the problem.
* **Strong Theoretical Results:** The paper provides a complete and rigorous theoretical analysis. Deriving the $O(T^{\frac{d}{d+2}}d^{\frac{d}{d+2}} + d\sqrt{T \log T})$ regret bound is a strong technical achievement. Proving its optimality by establishing a matching $\Omega(T^{\frac{d}{d+2}})$ lower bound makes the theoretical contribution self-contained and highly significant.
* **Solid Empirical Validation:** The experiments are well-designed and provide compelling support for the claims. The synthetic experiments directly validate the theoretical regret rate $\frac{d}{d+2}$, which is excellent. The real-world experiments on healthcare Q&A datasets convincingly demonstrate the algorithm's practical utility, showing it achieves a superior Pareto frontier for the cost-vs-mismatch-loss trade-off compared to a sensible baseline.
* **Clarity:** The paper is exceptionally well-written. The problem is motivated clearly with concrete examples, the formal problem setup is precise, and the algorithm's intuition is explained very well.

**Weaknesses:**

* **Gap Between Theory and Practical Implementation:** The strong theoretical guarantees (optimality, regret rate) are derived under the assumption of a quadratic loss $d(x,f) = (x-f)^\top W (x-f)$. This model has $O(d^2)$ parameters and leads to an "impractical" $O(d^4)$ complexity per-step, as the authors note. Consequently, the real-world experiments use computationally cheaper models: a rank-1 (squared linear) model $(\theta^\top(x-f))^2$ or a neural network. This creates a disconnect, as the theoretical guarantees do not strictly apply to the models used in the high-dimensional, practical validation. While the empirical results are strong, the paper would be improved by discussing this limitation more directly.
* **Scalability of the Action Library $S_t$:** The algorithm, as presented, requires computing the LCB for *every* existing context key $f \in S_t$ at *each* time step $t$. In a long-running system, the library $S_t$ could grow very large, potentially approaching $O(T)$ in size. This implies a per-step complexity that scales at least linearly with $|S_t|$ (e.g., $O(|S_t| \cdot D^2)$ for the NN model). The paper notes the *expected* number of created contexts is sublinear $O(T^{\frac{d}{d+2}})$, but this is not a worst-case guarantee. The paper would benefit from a discussion on the practical scalability for very large $T$ and whether mitigation strategies (e.g., approximate nearest-neighbor search for $f_t$, pruning the library $S_t$) would be necessary.
* **Baseline for Comparison:** The "Fixed-P" baseline is a reasonable "naive" strategy. However, it is arguably too simple. It decides to create *independently* of the context $x_t$, and when it reuses, it performs pure exploitation (picking the most similar context). A stronger baseline might involve running a standard contextual bandit algorithm (like LinUCB) on the *current* set $S_t$ (to intelligently explore/exploit *existing* actions) and pairing this with a more intelligent heuristic for creation (e.g., "create if the LCB loss of the best existing action is still greater than $c$"). Comparing against such a baseline would provide a clearer picture of the value added by the specific "doubly-optimistic" creation rule.

**Questions:**

1.  **On the Theory-Practice Gap:** Could you elaborate on the theoretical guarantees for the simplified models used in the real-world experiments (the squared linear model and the neural network)? Does the $O(T^{\frac{d}{d+2}})$ regret component (which seems related to the "facility location" aspect of the problem) still hold, even if the second regret term (from parameter estimation) changes from $O(d\sqrt{T \log T})$?
2.  **On Scalability:** How did the algorithm's runtime scale empirically with $T$ and the library size $|S_t|$ in your Q&A experiments? The per-step complexity seems to be $O(|S_t| \cdot \text{dim})$. Do you foresee scalability issues as $T$ becomes very large, and do you have recommendations for mitigating this (e.g., nearest-neighbor search, action pruning)?
3.  **On Baselines:** Could you comment on how your algorithm might compare to a baseline that uses a standard contextual bandit (like LinUCB) to select from $S_t$, paired with a deterministic creation rule like: "create a new action if and only if $\min_{f \in S_t} \tilde{d}_t(x_t, f) > c$"? This seems like a natural alternative, and I am curious why the stochastic UCB-based rule is preferable.
4.  **On the Oracle:** The analysis assumes the creation oracle $\mathcal{A}(x_t)$ is "perfect" and provides a zero-loss action. How do you think the problem and algorithm would need to change if the oracle was imperfect, i.e., generating a new action still incurred some unknown (but hopefully small) mismatch loss $l_t(\text{new}) > 0$, in addition to the fixed cost $c$?

---

> ### Author Response · Authors · 2025-11-18
>
> Thanks for your strong support and insightful feedback! We address your concerns and questions as follows:
>
> **1, Gap Between Theory and Practical Implementation:**
>
> Thank you for highlighting the theory–practice gap. We agree and will make it explicit. Our regret analysis is derived under a quadratic loss $d(x,f)=(x−f)ᵀW(x−f)$ with $W\succeq 0$, which is a tractable surrogate that captures mismatch in two common settings: (i) metric‑learning on semantic embeddings, where a learned W reweights dimensions to reflect task‑relevant similarity; and (ii) local second‑order approximations of excess risk $h(x, A(f))−h(x, A(x))$, whose Taylor expansion yields a quadratic form in $x−f$ with Hessian $H$.
>
> On theoretical analysis for practical models, we take two steps to match the gap. First, the rank-1 model (i.e. the squared linear loss) used in our experiments is a direct, computationally efficient specialization of the general quadratic model, where the matrix $W$ is constrained to be of the form $\theta\theta^\top$. The $O(T^{d/(d+2)}d^{d/(d+2)})$ term arises from the create‑to‑reuse geometry (a covering/facility‑location effect) and does not hinge on linear parametrization. Therefore, at least the $O(T^{d/(d+2)}d^{d/(d+2)} + d\sqrt{T\log T})$ regret *upper bound* still holds, although the second term is possible to be reduced to $d\log T$ as the MSE of linear regression if we *know* the rank-1 structure ahead of time. Second, in our high‑dimensional experiments, we keep the algorithmic principle (LCB for reuse, UCB‑based creation) and replace the estimator by using a neural-network, whose uncertainty is quantified by a Gaussian conjugate assumption. We have added a theoretical analysis for this uncertainty quantification in the Appendix C, which is not an end-to-end analysis of the regret guarantee as NN being involved, but is a proper derivation of the confidence bounds $Δ_t(x,f)$, enabling the doubly-optimistic principle to extend to neural approximators.
>
> We foresee the doubly-optimistic approach extending to broader Hölder-smooth and kernel-based loss classes through careful discretization and adaptive partitioning, though analyzing the regret bounds is a substantial future work.
>
> **2, Scalability of the Action Library:**
>
> We agree that computing an LCB for every $f∈S_t$ implies a per‑step cost linear in $|S_t|$. While our theory shows the expected number of created contexts scales as $O(T^{d/(d+2)})$, in high‑dimensional embeddings the exponent $d/(d+2)\rightarrow1$, so near‑linear growth can indeed occur for small $c$. To make this concrete, we added a discussion in Appendix A.2 along with a figure plotting $|S_t|$ versus $t$ for several fixed creation costs $c$. Empirically, the growth rate is strongly governed by $c$: larger $c$ yields much slower growth, which also matches with the results in Section 6.2. Furthermore, we agree with the excellent suggestions for mitigation and have included a detailed discussion of them in the same Appendix paragraph, which covers the approximate nearest-neighbor method as you suggested and a de-activation method that maintains a frequently-used action subset on top of the library.
>
> (to be continued in the next **Official Comment** block)

---

> ### Author Response · Authors · 2025-11-18
>
> (continuing from previous block)
>
> **3, New Baseline for Comparison:**
>
> We implemented the proposed variant of LinUCB (in our loss‑minimization setting, **LinLCB** over $S_t$) to select the best existing action, paired with a deterministic creation rule that creates iff $\min_f LCB_t(x_t,f) > c’$ and we swept the threshold $c’$ (which is different from the creation cost $c$) to plot the generation-mismatch tradeoff curve as we did in the main paper. The results are displayed and discussed in Appendix A.2.
>
> Empirically, this LinLCB baseline performs *worse* than our method, and it even goes above (worse than) the Fixed-P baseline frequently. The main reason is a coupling effect: using an LCB both to select among existing actions and to trigger creation makes creation excessively conservative in early rounds (LCBs are low under high uncertainty). This suppresses the addition of new actions, which in turn limits future exploration capacity and compounds the problem—leading to fewer candidates and higher mismatch later. It end up getting a negative feedback loop:
>
> (i) Optimism in LinLCB lowers the best available LCB.
>
> (ii) This suppresses the creation of new, necessary library items.
>
> (iii) The library $S_t$ remains small and insufficient, leading to poor learning and high modeling error in the long run.
>
> In contrast, the "Fixed-P" baseline, while naive, ensures a constant and stable rate of library growth, preventing this collapse into a state of insufficient learning. This result highlights the critical difference between “exploration” versus “creation” that might be misunderstood intuitively. To this extent, our “doubly-optimistic” framework doesn't just optimistically evaluate existing actions; it optimistically evaluates the choice to create a new action by treating $c$ as its potential value.
>
> In addition, we conjecture that the "fixed-threshold" strategy is similar to our Bernoulli random strategy under the i.i.d. assumption on the input covariates series, in terms of the Pareto front of generation-mismatch tradeoffs (although the choice of threshold $c’$ is different from the choice of $c$ to achieve the same balance). We adopt this randomized algorithm mainly for the simplicity of theoretical analysis.
>
> **4, On the Oracle:**
>
> This is an interesting question as it touches upon a critical and practical extension of our work. If the oracle is imperfect, creating a new action incurs not just a fixed cost $c$, but also an unknown mismatch loss $l_t(\text{new}) > 0$, this fundamentally changes the problem: Since our algorithm operates entirely in the representation space (observing $\{x_t\}$ and learning the mismatch function $d(x,f)$), we have no direct way to distinguish a "perfect" versus "imperfect" newly-created action. If the oracle is imperfect, the mismatch would manifest as bias or noise in the *observed* *representation*: when we request $\mathcal{A}(x_t)$, the oracle might actually generate $\mathcal{A}(x')$ for some $x' \neq x_t$, but we observe $x_t$ as a noisy indicator.
>
> The impact on regret analysis would heavily depend on the structure and assumptions of oracle imperfection: Ideally, if the error is i.i.d. and sub-Gaussian, it may affect the hindsight optimal sequence of decisions ($OPT_h$) and the algorithm symmetrically, potentially preserving the regret rate (though with worse constants). However, our current analysis benchmarks against $OPT_o$ that sets optimal actions according to the underlying covariate distribution ahead of time. Since $OPT_o$ is not dependent on the sequential observations, the gap between $OPT_o$ and $OPT_h$ could accumulate to $O(T)$ under persistent oracle error caused by mismatching, therefore fundamentally changing the problem and the analysis. We believe this is a very promising direction for future work.
>
> --------------------------------
>
> We hope these have addressed your concerns. Please let us know if there are any other questions. Thanks!

---

### Author Response · Authors · 2025-11-18
**Overall Response to Reviewers**

We thank all reviewers for their thoughtful comments and positive feedback! We have carefully addressed each concern and made the following revisions to strengthen the paper:

**1, Growth of Action Set Size (Appendix A.2):** We have added a new experiment-based discussion analyzing the growth of the action set size $|S_t|$ over the time horizon $T$ for different choices of the creation cost $c$. This demonstrates that our algorithm adaptively controls library growth—higher costs lead to slower accumulation and more reuse, validating the economic tradeoff at the core of our framework.

**2, LinLCB Baseline Comparison (Appendix A.2):** We include a comparison against a LinLCB (a version of LinUCB in terms of loss minimization) baseline that uses a deterministic threshold for creation decisions (instead of our probabilistic UCB/c rule). The results show that our doubly-optimistic decision mechanism achieves superior cost-effectiveness by better balancing creation versus mismatching in expectation.

**3, Neural Network Confidence Bounds (Appendix C):** We provide a formal derivation of the confidence bounds $\Delta_t(x,f)$ for neural network distance estimators using Laplace approximation of the Gaussian posterior. This justifies the doubly-optimistic principle (LCB-reuse, UCB-creation) we adopt in our high-dimensional Healthcare Q&A experiments, bridging methodology and implementation.

We are grateful for the opportunity to improve our work based on the reviewers' suggestions and believe these additions have made the paper stronger. Thank you again for your constructive guidance!

------------
Best,

Submission 14215 Authors

---

### Meta-Review · Area_Chair_qk3M · 2026-01-02

**Summary:**

All the Reviewers are positive about the paper. The concerns raised by the Reviewers have been adequately addressed by the Authors in the rebuttals. Thus, I recommend acceptance of this paper.

**Reviewer Concerns:**

While positive about the paper, the Reviewers raised some concerns about it. Specifically, the main concerns were: (i) a discussion on how the size of the action set grows as a function of the creation cost was missing, (ii) a comparison with LinUCB was needed, and (iii) a formal derivation of the confidence bounds for neural network distance estimators was not provided but needed. The Authors adequately addressed all these concerns in the rebuttals, by properly updating the paper.

**Reviewer Scores:**

Reviewer uKET, Score: 8 - I believe that the Reviewer's score would not have changed.

Reviewer WJ1K, Score: 8 - I believe that the Reviewer's score would not have changed.

Reviewer yeWb, Score: 6 - I believe that the Reviewer's score would not have changed.

Reviewer pGC5, Score: 6 - I believe that the Reviewer's score would not have changed.

Reviewer GG79, Score: 6 - I believe that the Reviewer's score would not have changed.

---

### Decision · Program_Chairs · 2026-01-26

Accept (Poster)